# Perturbational phenotyping of human blood cells reveals genetically determined latent traits associated with subsets of common diseases

Max Homilius [1,2,4,5] ✉, Wandi Zhu [1,2,4,5] ✉, Samuel S. Eddy[1], Patrick C. Thompson [1], Huahua Zheng[1], Caleb N. Warren[1], Chiara G. Evans[1], David D. Kim [1], Lucius L. Xuan [1], Cissy Nsubuga [1], Zachary Strecker[1], Christopher J. Pettit[1], Jungwoo Cho[1], Mikayla N. Howie[1], Alexandra S. Thaler[1], Evan Wilson [1], Bruce Wollison[1], Courtney Smith[1], Julia B. Nascimben [1], Diana N. Nascimben[1], Gabriella M. Lunati[1], Hassan C. Folks[1], Matthew Cupelo[1], Suriya Sridaran[1], Carolyn Rheinstein[1], Taylor McClennen [1], Shinichi Goto [1,2], James G. Truslow [1], Sara Vandenwijngaert[1], Calum A. MacRae [1,2,5] ✉ & Rahul C. Deo [1,2,3,5] ✉

Although genome-wide association studies (GWAS) have successfully linked genetic risk loci to various disorders, identifying underlying cellular biological mechanisms remains challenging due to the complex nature of common diseases. We established a framework using human peripheral blood cells, physical, chemical and pharmacological perturbations, and flow cytometry-based functional readouts to reveal latent cellular processes and performed GWAS based on these evoked traits in up to 2,600 individuals. We identified 119 genomic loci implicating 96 genes associated with these cellular responses and discovered associations between evoked blood phenotypes and subsets of common diseases. We found a population of pro-inflammatory anti-apoptotic neutrophils prevalent in individuals with specific subsets of cardiometabolic disease. Multigenic models based on this trait predicted the risk of developing chronic kidney disease in type 2 diabetes patients. By expanding the phenotypic space for human genetic studies, we could identify variants associated with large effect response differences, stratify patients and efficiently characterize the underlying biology.

Precision medicine strives to reclassify complex heterogeneous diseases into distinct biologically defined groups, thereby enabling targeted therapies and improved outcomes. Examples include the subdivision of common cancers by somatic driver mutations[1], the discovery of eosinophilic variants of asthma[2] and the recognition that some presentations of heart failure may arise from the accumulation of amyloidogenic proteins, which can be subdivided further based on the aggregating protein[3]. The realization of precision medicine has

[1]One Brave Idea and Division of Cardiovascular Medicine, Department of Medicine, Brigham and Women's Hospital, Boston, MA, USA. [2]Harvard Medical School, Boston, MA, USA. [3]Present address: Atman Health Inc, Needham, MA, USA. [4]These authors contributed equally: Max Homilius, Wandi Zhu. [5]These authors jointly supervised this work: Max Homilius, Wandi Zhu, Calum A. MacRae, Rahul C. Deo. ✉e-mail: mhomilius@bwh.harvard.edu; wzhu5@bwh.harvard.edu; cmacrae@bwh.harvard.edu; rdeo@bwh.harvard.edu

**Fig. 1 | Chemical perturbations expand the phenotypic space of quantitative blood profiles, and blood cell responses are associated with clinical phenotypes and genetic variants. a**, Recruitment setting and application of a standard hematological analyzer for CBC together with perturbation agents to systematically measure cellular responses in whole-blood samples across a clinical cohort. **b**, Data-driven gating strategy for four Sysmex channels, including WDF, WNR, PLT-F, and RET channels. Gates were defined according to known and new cellular states in response to perturbation conditions. **c**, Cell gates were used to derive high-dimensional quantitative readouts for 278 blood cell parameters across 37 environmental conditions including inflammatory stimuli (LPS and Pam3CSK4), heat or approved and experimental

compounds (dapagliflozin, empagliflozin and captopril). Each perturbation condition was measured for up to 3,300 individuals (see Supplementary Table 1 for a description of conditions and Extended Data Fig. 3 for a projection of blood-response readouts). **d**, Blood parameters and response to perturbation conditions were associated with clinical phenotypes such as ICD10 diagnostic codes and lab measurements. **e**, The perturbation screening setting yielded many genetic associations that were specific to blood cell types and environmental stimuli. By comparing similar conditions in the same cohort, detailed comparisons between perturbation conditions and specific associated blood parameters were possible. LPS, lipopolysaccharide; QTc, corrected QT interval.

been hindered by the lack of readily available measures of the activities of discrete biological pathways in most common diseases. Historical approaches have focused on mining large patient biobanks combining archived DNA, RNA and serum or plasma samples with clinical records[4]. Although such strategies have identified common genetic variants associated with clinical outcomes, they have typically not been successful at capturing the underlying cell biology, limiting their utility in producing mechanistic insights into therapeutic implications[5,6].

We aimed to establish a framework that bridges genetic variants and complex diseases through standardized phenotyping of primary human cells. We used live human blood cells, as these

reflect physiological processes, disease states and environmental factors, including active therapies. For example, dysregulation of hematopoietic processes can result in disease progression via mechanisms such as the contribution of inflammation to atherosclerosis and insulin resistance[7–9] or hyperactive coagulation in pathological thrombosis[10–12]. In addition to circulating cells with their repertoire of responses, blood plasma contains hormones, secreted proteins, metabolites, cell-free DNA, microparticles and extracellular vesicles that can carry signals to blood cells or other cell types. Peripheral blood may offer a diagnostic window into multiple organ systems and integrative physiology[13–15].

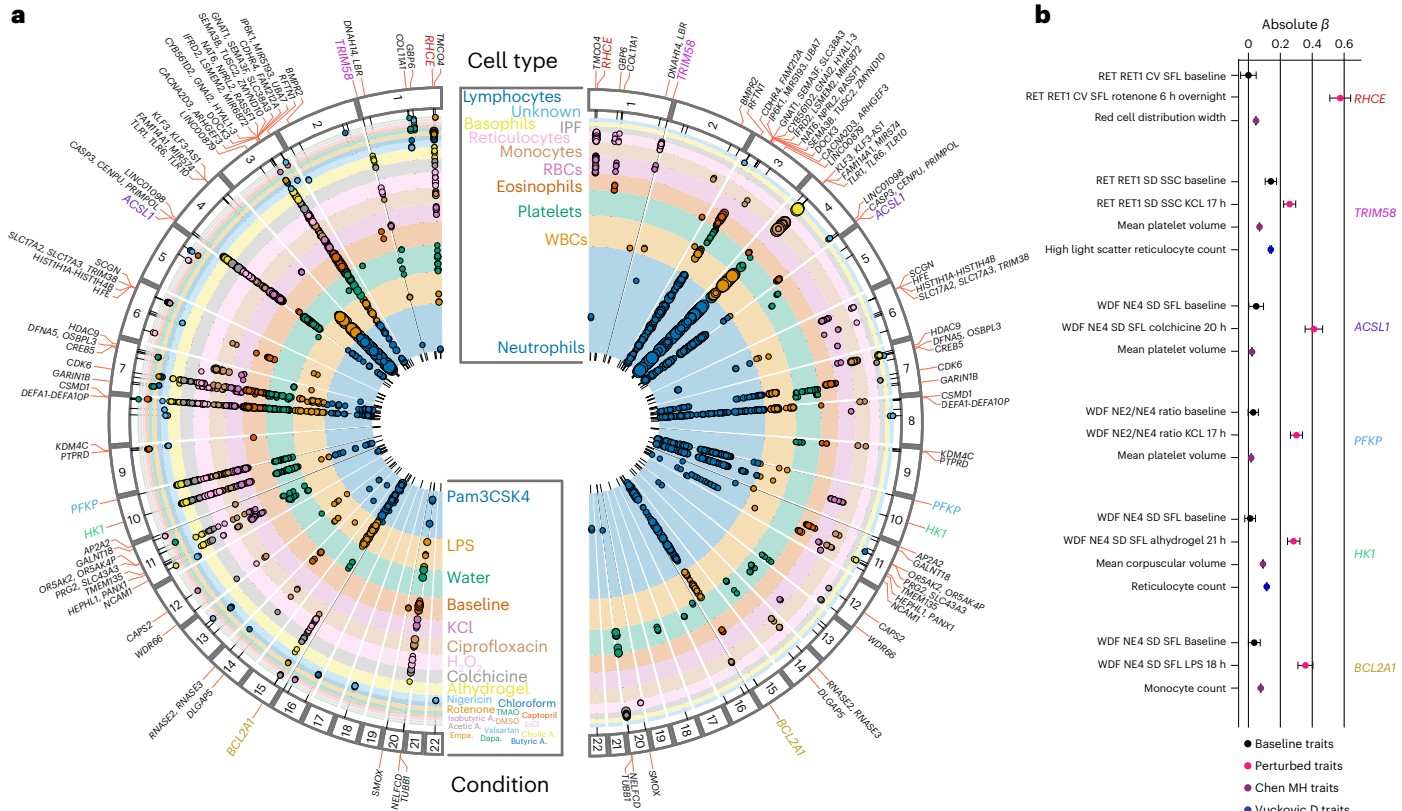

**Fig. 2 | Whole-blood perturbational profiling yields a wide range of genetic associations for specific conditions and cell types. a**, Genome-wide significant associations with $P < 5 \times 10^{-8}$ colored by perturbation condition (left) and cell type (right). Two-sided $P$ values are based on $t$ tests in linear regression models and are not adjusted for multiple testing. Circle size is proportional to $-\log_{10}$ ($P$ value). Nearby genes are annotated based on proximity. For clarity, only a subset of readouts is shown for loci with many significant associations (see Table 1 for an overview of traits, cell types, candidate genes and previously reported blood-trait associations and Supplementary Data 1 for a full listing

of associations). **b**, Comparison of $\beta$ coefficients for six of the most significant variants across multiple traits and genes. For these readouts, perturbation conditions led to large effect size changes that were not observed at baseline. For our study, the variants shown are rs644592 (*RHCE*, $n = 943$), rs3811444 (*TRIM58*, $n = 1,410$), rs12513029 (*ACSL1*, $n = 1,296$), rs34538474 (*PFKP*, $n = 1,339$), rs6480404 (*HK1*, $n = 1,378$) and rs67760360 (*BCL2A1*, $n = 1,424$). For the studies in refs. 17,18, which included over 400,000 individuals, the variants shown are the reported variants with the lowest $P$ value for each gene. Data are presented as absolute estimated $\beta$ coefficient ±s.e.m.

Previous genome-wide association studies (GWAS) on whole blood primarily focused on complete blood counts (CBCs); clinical parameters describing numbers; volumes and distribution of leukocytes; erythrocytes and platelets; and the genetic architecture of hematopoiesis and blood diseases have been mapped in detail[16–18]. A recent study expanded measured phenotypes to include flow cytometry-derived parameters with the aim of better describing cellular function[19]. The Human Functional Genomics Project profiled cytokine production and baseline immune parameters in response to pathogen challenges[20]. Other studies have revealed the genetic basis of platelet aggregation in response to known agonists[21,22]. However, these studies did not consider the dynamic responses of blood cells to environmental conditions, which likely contribute to their effects on disease development, progression and prevention.

We hypothesized that treating whole blood ex vivo with diverse stressors or stimuli would enable the identification of latent differential cellular responses and new disease-associated endophenotypes. We anticipated that this expansion of phenotypic space would evoke traits determined by large effect size common alleles, enabling efficient target identification and improving the prediction of incident events. Moreover, given that biological pathways are reused across diverse tissues and organ systems, insights into whole blood may be relevant to a range of conditions originating in different tissues. By identifying intermediate cellular phenotypes, we sought to define subcategories

of disease and specific pathophysiologic mechanisms that can be targeted more directly.

## Results

### Chemical perturbations expand the phenotypic space of blood profiles

In clinical settings, whole blood cytometry is used to quantify circulating cells as part of standardized diagnostic tests. We adapted a widely-used whole-blood cytometry analyzer (Sysmex XN-1000) to systematically profile peripheral blood from over 4,700 study participants (donors) under 37 conditions (36 perturbations and baseline), genotyped more than 2,600 donors and performed GWAS for all blood perturbation profiles (Fig. 1a). We recorded side scatter (SSC), forward scatter (FSC) and side fluorescence (SFL) of blood cells using four fluorescence dyes (white cell differential channel by fluorescence (WDF), white count and nucleated red blood cells (WNR), reticulocyte (RET) and platelet F (PLT-F)) that quantify morphological and intracellular properties. Chemical stressors evoked distinct cellular states for blood cells that were not typically observed under baseline conditions, enabling the detection of new cell populations in three-dimensional cytometry measurements (Extended Data Fig. 1). We determined cellular gates based on empiric distributions of blood cells under perturbation conditions and defined parameter sets for all observed cell populations (Fig. 1b and Extended Data Fig. 2). The perturbation conditions represented discrete

**Table 1 | Genetic regions associated with whole-blood perturbation response traits**

| Lead SNP | rsID | P value | Candidate genes | CADD consequence | Top trait | Obs. | Previous association |
|---|---|---|---|---|---|---|---|
| 1:20032226:G:A | rs10917522 | $3.09 \times 10^{-9}$ | TMCO4 | Intron | WDF Empa 1.5 h NE3 CV SFL | 380 | – |
| 1:25703156:C:T | rs644592 | $5.58 \times 10^{-18}$ | RHCE | Intron | RET rotenone 6 h ov. RET1 CV SFL | 943 | RBC[a] |
| 1:89840389:T:C | rs7550358 | $1.55 \times 10^{-8}$ | GBP6 | Intron | RET captopril 5.5 h RET2 Count | 353 | – |
| 1:103361529:A:C | rs72683260 | $3.22 \times 10^{-9}$ | COL11A1 | Intron | RET TMAO 3.5 h RBC2 Med SFL | 361 | – |
| 1:225579918:A:T | rs41268717 | $6.90 \times 10^{-9}$ | DNAH14, LBR | Intron | WNR water 15 h WBC2 Med FSC | 1,423 | RBC[a] |
| 1:248039451:C:T | rs3811444 | $1.37 \times 10^{-11}$ | TRIM58 | Missense | RET KCl 17 h RET1 SD SSC | 1,410 | RBC[a], PLT[b] |
| 2:203226371:G:A | rs72925015 | $1.24 \times 10^{-8}$ | BMPR2 | Upstream | WDF water 15 h MO2 Med SSC | 1,392 | RBC[a] |
| 3:16551213:C:G | rs2881513 | $3.78 \times 10^{-8}$ | RFTN1 | Regulatory, intron | WNR nigericin 0.5 h UK1 CV FSC | 327 | – |
| 3:49774658:G:A | rs73077175 | $1.01 \times 10^{-13}$ | CDHR4-UBA7, IP6K1 | Intron | WDF baseline NE2 Med SFL | 1,629 | RBC[a] |
| 3:50255663:C:T | rs35926495 | $8.32 \times 10^{-25}$ | SLC38A3 | Intron | WDF baseline NE2 Med SFL | 1,664 | RBC[a] |
| 3:50374293:A:G | rs2073499 | $9.63 \times 10^{-9}$ | HYAL3, RASSF1 | Regulatory, intron | WDF baseline NE2/NE4 ratio | 1,565 | BASO[a] |
| 3:51406862:A:G | rs111614418 | $2.29 \times 10^{-8}$ | DOCK3 | Intron | WNR LPS 18 h WBC Med SSC | 1,416 | EO[b] |
| 3:56849749:T:C | rs1354034 | $7.23 \times 10^{-10}$ | ARHGEF3 | Intron | RET KCl 17 h PLT Med SFL | 1,397 | PLT, RBC, LY[b] |
| 3:94702472:C:T | rs1432474 | $1.92 \times 10^{-8}$ | LINC00879 | Intron | WDF water 23 h MO2 CV FSC | 1,415 | – |
| 4:38677227:A:C | rs34089598 | $7.94 \times 10^{-12}$ | KLF3, KLF3-AS1 | Regulatory, intron | WNR Pam3CSK4 19 h WBC CV FSC | 1,310 | WBC[b] |
| 4:38798648:C:A | rs5743618 | $8.20 \times 10^{-103}$ | TLR1, TLR6, TLR10 | Missense | WDF Pam3CSK4 19 h NE1 Med FSC | 1,300 | – |
| 4:178716833:T:C | rs10030190 | $4.08 \times 10^{-8}$ | LINC01098 | Intron | WNR baseline UK1 CV FSC | 1,486 | – |
| 4:185602707:G:A | rs72703519 | $2.92 \times 10^{-20}$ | CASP3-ACSL1 | Intron | WDF KCl 17 h NE2/NE4 ratio | 1,336 | – |
| 4:185665118:G:A | rs12513029 | $1.55 \times 10^{-13}$ | CASP3-ACSL1 | Intergenic | WDF colchicine 20 h NE4 SD SFL | 1,296 | PLT[a] |
| 6:25719210:T:C | rs9358870 | $3.71 \times 10^{-9}$ | SCGN | Intergenic | RET DMSO 4.5 h RBC1 SD FSC | 355 | PLT[b] |
| 6:25878848:A:G | rs55925606 | $2.97 \times 10^{-9}$ | HFE-TRIM38 | Upstream and downstream | RET DMSO 4.5 h RBC1 CV FSC | 381 | RBC, PLT[b] |
| 7:18398911:C:T | rs62450075 | $9.82 \times 10^{-9}$ | HDAC9 | Intron | RET KCl 17 h RBC1 SD FSC | 1,381 | – |
| 7:24832308:A:G | rs4719781 | $2.50 \times 10^{-18}$ | DFNA5, OSBPL3 | Downstream | WNR ciprofloxacin 22 h BASO Med SSC | 1,260 | – |
| 7:28773957:A:C | rs73075771 | $1.19 \times 10^{-8}$ | CREB5 | Intron | WNR TMAO 3.5 h UK1 CV SSC | 325 | WBC[b] |
| 7:92408370:C:T | rs445 | $2.30 \times 10^{-14}$ | CDK6 | Regulatory, intron | WDF baseline EO1 Med SSC | 1,698 | WBC, RBC[b] |
| 7:128371246:C:T | rs41274144 | $6.64 \times 10^{-9}$ | GARIN1B | 3' UTR | WNR TMAO 3.5 h PLT CV SSC | 327 | – |
| 8:4096691:T:C | rs28522529 | $2.87 \times 10^{-10}$ | CSMD1 | Intron | WDF captopril 5.5 h MO2 CV SFL | 343 | – |
| 8:6828115:G:T | rs2615764 | $1.89 \times 10^{-17}$ | DEFA10P | Upstream | PLT-F baseline WBC1 Med SSC | 1,662 | WBC[b] |
| 9:7015133:A:G | rs10975974 | $3.39 \times 10^{-10}$ | KDM4C | Intron | WDF baseline MO2 Med SSC | 1,688 | RBC[a] |
| 9:9744225:A:C | rs80353904 | $3.10 \times 10^{-8}$ | PTPRD | Intron | WDF nigericin 7.5 h EO2 CV SSC | 351 | – |
| 10:3139540:A:G | rs34538474 | $6.55 \times 10^{-15}$ | PFKP | Intron | WDF KCl 17 h NE2/NE4 ratio | 1,339 | PLT[a] |
| 10:71109406:T:C | rs6480404 | $4.03 \times 10^{-13}$ | HK1 | Regulatory, intron | WDF Alhydrogel 21 h NE4 SD SFL | 1,378 | RBC[b] |
| 11:972270:C:T | rs7933889 | $1.03 \times 10^{-8}$ | AP2A2 | Intron | WNR ciprofloxacin 22 h WBC2 SD SFL | 1,358 | – |
| 11:11548147:A:G | rs10831631 | $3.19 \times 10^{-9}$ | GALNT18 | Intron | WDF LiCL 4 h NE1 CV FSC | 369 | – |
| 11:56806558:C:T | rs12421419 | $4.11 \times 10^{-9}$ | OR5AK2, OR5AK4P | Downstream | WDF colchicine 20 h LY SD SSC | 1,338 | – |
| 11:57159189:T:C | rs548854 | $1.81 \times 10^{-12}$ | PRG2, SLC43A3 | Upstream, intron | WDF colchicine 20 h EO1 Med FSC | 1,383 | – |
| 11:87048905:G:A | rs4536247 | $9.81 \times 10^{-9}$ | TMEM135 | Intergenic | WDF water 15 h NE2% | 1,358 | – |

**Table 1 (continued) | Genetic regions associated with whole-blood perturbation response traits**

| Lead SNP | rsID | *P* value | Candidate genes | CADD consequence | Top trait | Obs. | Previous association |
|---|---|---|---|---|---|---|---|
| 11:93862020:C:T | rs4753126 | $3.58 \times 10^{-12}$ | *HEPHL1, PANX1* | Regulatory, upstream | WDF colchicine 20 h EO2 Med SFL | 1,319 | RBC[a] |
| 11:112971545:C:T | rs11214488 | $2.16 \times 10^{-8}$ | *NCAM1* | Intron | WDF cholic acid 6.5 h NE3 CV SSC | 360 | – |
| 12:75695577:A:G | rs10785185 | $2.62 \times 10^{-8}$ | *CAPS2* | Intron | PLT-F isobutyric 3 h IPF SD FSC | 370 | – |
| 12:122399173:C:A | rs11615667 | $1.24 \times 10^{-9}$ | *WDR66* | Intron | PLT-F ciprofloxacin 22 h IPF SD SFL | 1,284 | PLT[a] |
| 14:21347966:G:T | rs74034667 | $1.88 \times 10^{-10}$ | *RNASE3* | Upstream | WDF baseline MO2 CV SFL | 1,700 | – |
| 14:21423790:G:C | rs2013109 | $8.60 \times 10^{-12}$ | *RNASE2* | Intron | WDF baseline MO CV SFL | 1,651 | – |
| 14:55654183:T:C | rs2094103 | $1.01 \times 10^{-8}$ | *DLGAP5* | Intron | PLT-F ciprofloxacin 22 h PLT-F SD FSC | 1,399 | – |
| 15:80260872:G:A | rs67760360 | $6.95 \times 10^{-21}$ | *BCL2A1* | Regulatory, intron | WDF LPS 18 h NE4 CV SFL | 1,430 | WBC[b] |
| 20:4157072:C:G | rs6084653 | $3.94 \times 10^{-10}$ | *SMOX* | Intron | RET baseline RET2 CV SFL | 1,605 | RBC[b] |
| 20:57569860:C:G | rs1043219 | $4.09 \times 10^{-10}$ | *NELFCD* | Downstream, 3'UTR | RET colchicine 20 h PLT CV SFL | 1,334 | PLT[b] |
| 20:57597970:A:C | rs463312 | $1.19 \times 10^{-19}$ | *TUBB1* | Missense, downstream | PLT-F baseline IPF SD SFL | 1,681 | PLT[b] |

Associations for blood traits and perturbation conditions were clumped to produce unique genomic regions across multiple conditions. Two-sided *P* values are based on *t* tests in linear regression models and are not adjusted for multiple testing. Variants with the lowest *P* value for each clumped region were selected as lead SNPs. The trait names contain the channel, condition and readout; for example, WDF Empa 1.5 h NE3 CV SFL indicates a readout in the WDF channel, with empagliflozin treatment, quantifying the SFL CV of a neutrophil subpopulation (NE3). This table contains a subset of regions with nearby candidate genes (see Supplementary Data 1 for a complete listing of associations). CV, coefficient of variation. [a]The previous association identified in ref. 17, which analyzed over 560,000 individuals. [b]The previous association identified in ref. 16, which analyzed over 173,000 individuals.

classes of exposure likely to contribute to blood cell responses as follows: (1) simulated physiological stressors; (2) chemical stressors; (3) gut microbiome metabolites; and (4) drugs with known mechanisms of action (Supplementary Table 1). We recorded up to 37 condition-specific blood responses for each donor and calculated quantitative profiles characterizing each cell population using cell counts, as well as median and s.d. for SSC, FSC and SFL parameters for each blood cell population (Fig. 1c and Supplementary Table 2). Compared to the baseline, each perturbation evoked particular changes in the characteristics of different blood lineages, resulting in a series of distinct cellular profiles (Extended Data Figs. 1 and 2 and Supplementary Fig. 1). With these chemical perturbations, we expanded quantification for each donor from 278 parameters to more than 4,000 parameters on average, greatly expanding the phenotypic space that could be interrogated.

Across the 36 perturbations, we collected measurements from 650 to 3,300 donors per condition. We then associated blood-response profiles with clinical traits, including quantitative lab values and diagnostic codes, to identify clinical endpoints and disease syndromes reflected in the evoked blood-response readouts (Fig. 1d). We also identified genetic loci associated with blood perturbation responses, which were often specific to perturbation conditions, cell populations and physical readouts (Fig. 1e). When comparing blood-response profiles, the perturbation conditions, readouts and associated genetic loci formed clusters of related conditions and cell types (Extended Data Fig. 3), suggesting the evoked blood profiles are informative for specific biological processes.

**Perturbational conditions yield new genetic associations**

To determine genetic variants associated with perturbation blood cell responses, we tested linear, univariate associations of 278 cellular phenotypes in 37 different conditions against >3.5 million imputed variants in 260–2,200 donors. We clumped variants with high linkage disequilibrium (LD) to identify more than 100 genomic loci that were significantly associated with at least two blood perturbation readouts (Supplementary Data 1). We identified 48 unique, nonoverlapping regions with nearby candidate genes (Fig. 2a and Table 1). Approximately half of the identified associations (25 of 48 genetic regions with

candidate genes) had previously been described as blood biomarker associations under baseline conditions with parameters that are part of CBC studies encompassing 170,000 to over 700,000 individuals[16–18]. We observed new associations in previously unreported cell types for many previously reported loci (12 of 25), such as white blood cell (WBC) responses associated with *SLC83A3*, whereas only RET-based associations had previously been described[17]. Additionally, we identified 23 new regions associated with blood cell responses to perturbations that have not been described, for example, the response to empagliflozin associated with variants in *TMCO4*. This gene had previously been associated with chronic inflammatory diseases[23]. Most associations we observed were specific to a particular blood lineage, such as RET readouts associated with *TRIM58* or neutrophil-specific associations with *PFKP* and *ACSL1*.

Chemical stressors increased response differences among donors (Extended Data Figs. 1 and 2 and Supplementary Fig. 1), making it possible to identify robust genetic associations with small sample sizes. For example, neutrophil and other WBC responses induced by inflammatory stimuli such as Pam3CSK4 or lipopolysaccharide (LPS) showed strong associations with a missense variant in *TLR1* (for example, rs5743618, WDF_Pam3CSK4_19h_NE1_Med_FSC; $P = 8.2 \times 10^{-103}$, $n = 1,300$). This association between *TLR1* and WBC traits was not described previously in cohorts studying CBC parameters with over half a million individuals. The same SNP has previously been associated with asthma and allergic diseases through unclear mechanisms[24]. Our results suggest a potential role for neutrophils as mediators in these disease phenotypes. Comparing $\beta$ coefficients for six genes that were previously identified in blood-trait GWAS showed that perturbation conditions greatly increased observed effect sizes compared to baseline conditions (Fig. 2b).

**Blood perturbation responses reflect organ-specific disease traits**

To assess whether perturbation-based blood cell traits reflect individuals' disease status, we tested for associations between 327 blood readouts (top three traits with the lowest GWAS *P* value were selected for each unique locus) and a collection of structured phenotypes based

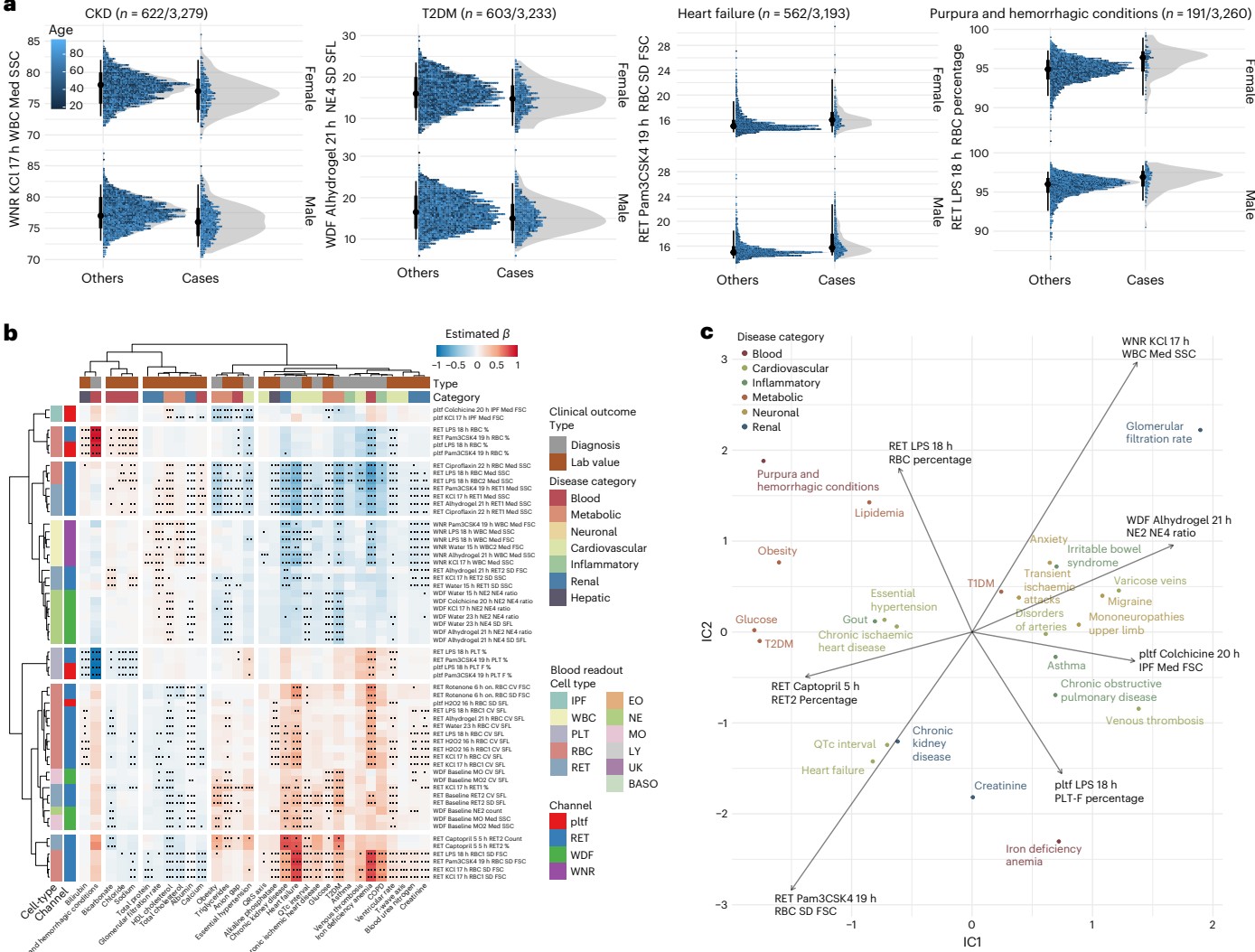

**Fig. 3 | Blood readouts under perturbation conditions are associated with clinical traits. a**, Distribution of raw blood readouts with associated clinical diagnoses. Each point shows readout for one study participant, stratified by sex and disease status, with color indicating age at blood draw. The gray area illustrates the normalized density of readouts for each subgroup. **b**, Pairwise association between quantile-transformed blood readouts and clinical lab values or diagnostic codes. Association effect sizes were estimated using linear and logistic regression models for quantitative lab measurements and binary traits, respectively. Positive associations are shown in red; negative associations are shown in blue. Only associations for a subset of blood traits are shown (see Supplementary Fig. 2 for all blood traits that have significant genetic associations). $P$ values were adjusted for FDR to account for multiple testing

across 327 perturbational blood readouts and 50 clinical outcomes, including 20 lab values and 30 diagnoses. Points indicate significant associations with adjusted $P$ value thresholds as follows: one point signifies 0.001, two points signify 0.001 and three points signify 0.0001 (see Supplementary Table 4 for clinical trait definitions using diagnostic codes and Supplementary Data 2 for all association results with FDR < 0.1). **c**, ICA of the $t$ score matrix of associations between blood readouts and clinical endpoints. Shown is a subset of diagnoses and lab values projected onto the first two independent components together with mixing matrix loadings of selected blood readouts. FDR, false discovery rate; ICA, independent component analysis; T2DM, type 2 diabetes mellitus; T1DM, type 1 diabetes mellitus.

on electronic health record (EHR) data. Diagnostic status for multiple common disorders was significantly associated with variation in blood perturbation readouts (Fig. 3a,b, Supplementary Fig. 2 and Supplementary Data 2). Notably, perturbations elicited unique disease associations absent at baseline. For example, neutrophil variability in SFL at baseline (WDF_Baseline_NE4_SD_SFL) showed no significant association with disease. However, the same parameter under 21 h Alhydrogel perturbation (WDF_Alhydrogel_21h_NE4_SD_SFL) showed negative associations with multiple cardiometabolic diseases, including heart failure (cases = 532, $t = -2.98$, $P_{adj} = 0.014$), type 2 diabetes (T2D; cases = 685, $t = -6.43$, $P_{adj} = 5.4 \times 10^{-9}$) and chronic kidney disease (CKD; cases = 546, $t = -3.48$, $P_{adj} = 3.45 \times 10^{-3}$). Certain blood readouts showed associations

with very specific disease phenotypes; for example, platelet variability in SFL under KCl 17 h perturbation was positively associated with purpura and hemorrhagic conditions (RET_KCl_17h_PLT_CV_SFL: cases = 225, $t = 8.16$, $P_{adj} = 4.89 \times 10^{-14}$) and negatively associated with venous thrombosis (RET_KCl_17h_PLT_CV_SFL: cases = 220, $t = -3.78$, $P_{adj} = 1.3 \times 10^{-3}$).

In addition to diagnostic codes, quantitative lab values commonly used to assess various physiological parameters also demonstrated robust associations with blood perturbation responses. For example, red blood cell (RBC) median SSC under 18 h LPS condition (RET_LPS_18 h_RBC_Med_SSC) showed strong positive associations with serum albumin ($n = 2,494$, $t = 11.75$, $P_{adj} = 3.89 \times 10^{-28}$) and eGFR

($n = 2,569$, $t = 3.26$, $P_{adj} = 6.66 \times 10^{-3}$), which corresponds with its negative association with CKD status. Significant associations included clinical traits that are not directly measurable in blood, such as a positive correlation between corrected QT interval on an electrocardiogram and RBC size variability under 18-h LPS perturbation (RET_LPS_18h_RBC1_SD_FSC; $n = 1,946$, $t = 10.22$, $P_{adj} = 3.01 \times 10^{-21}$), indicating that latent blood phenotypes may reflect physiological changes occurring in other tissues.

To explore the associations between blood traits and clinical phenotypes, we employed independent component analysis (ICA) to identify maximally uncorrelated components in the association matrix (Fig. 3c). ICA effectively grouped clinical endpoints and lab values into meaningful clusters, for example, one encompassing obesity, T2D and glucose measurements, and another comprising asthma, chronic obstructive pulmonary disease and venous thrombosis (Fig. 3c). We plotted the loadings of seven example blood traits onto the same IC space (Fig. 3c, arrows), demonstrating how each blood trait carries unique information related to clinical phenotypes. We found that many perturbation conditions elicited new clinical associations not observed at baseline, suggesting perturbations evoked unique previously latent blood cell responses that are disease-relevant.

## A neutrophil population is negatively associated with cardiometabolic phenotypes

Multiple chemical stimuli, when studied with long exposure durations, elicited a distinct population of neutrophils (NE2) in the Sysmex WDF channel, exhibiting high SSC and fluorescence measurements, which were absent under baseline conditions (Fig. 1b). As an exemplar, we investigated this phenotype and functionally characterized this neutrophil population.

The ratio of NE2 neutrophils to the total neutrophil count (NE2/NE4) under multiple chemical perturbations showed associations with a complex aggregate of cardiometabolic diseases, specifically chronic ischemic heart disease, heart failure and T2D. For example, the NE2/NE4 ratio with an inflammatory stimulus (WDF_Pam3CSK4_19h_NE2/NE4) had negative associations with T2D (cases = 685, $t = -5.4$, $P_{adj} = 1.51 \times 10^{-6}$), obesity (cases = 1,202, $t = -4.37$, $P_{adj} = 1.47 \times 10^{-4}$) and related lab values (serum triglycerides: $n = 2,248$, $t = -5.6$, $P_{adj} = 7.64 \times 10^{-7}$; serum glucose: $n = 2,657$, $t = -3.7$, $P_{adj} = 1.72 \times 10^{-3}$). This blood readout also exhibited a positive correlation with total high-density lipoprotein cholesterol levels ($n = 2,259$, $t = 6.32$, $P_{adj} = 1.24 \times 10^{-8}$). These results suggest that a low NE2/NE4 ratio is associated with cardiometabolic disease phenotypes.

## The NE2 population represents apoptotic neutrophils

Because the NE2 population was only observed with perturbations at later time points, we hypothesized that it was related to neutrophil death. To evaluate the biological processes occurring in this population, we developed protocols to label purified neutrophils with the Sysmex WDF dye to visualize NE2 using regular flow cytometry (Fig. 4a). We found that the cells that represent the NE2 population, showing elevated WDF dye fluorescence and SSC, exhibited increased signals in Annexin V and Sytox, compared to the NE1-like population that mirrors the normal neutrophil profile at baseline (Fig. 4b,c). Annexin V is a marker for early apoptosis, while Sytox is indicative of cell death. Furthermore, we observed that blood samples with higher NE2/NE4 ratios exhibited higher percentages of Sytox and Annexin V-positive neutrophils (Fig. 4d). These results suggest that the NE2 population elicited by various chemical perturbations represents a subset of neutrophils actively undergoing apoptosis.

## Pro-inflammatory responses delay neutrophil apoptosis

Delayed apoptosis and impaired clearance of neutrophils can lead to non-resolving inflammation and subsequent tissue damage[25,26]. Neutrophils have short lifespans[27], which can be prolonged by pro-inflammatory and pro-survival signals[26]. Patients with aggregated cardiometabolic diseases exhibited a decreased NE2/NE4 ratio, suggesting reduced neutrophil apoptosis. We hypothesized that the reduction in neutrophil apoptosis results from their increased pro-inflammatory responses. We examined neutrophil activation and generation of reactive oxygen species (ROS) at an earlier time point (4.5 h post-treatment) that is within the normal range of neutrophil half-life in vivo and compared it with the Sysmex NE2/NE4 readout at a later time point (17 h post-treatment; Fig. 4e). Neutrophil activation has been previously associated with the upregulation of CD11b on the cell membrane and shedding of CD62L[28,29]. Using these two surface markers, three distinct neutrophil populations are defined, such as CD11b$^{high}$ CD62L$^{low}$, CD11b$^{medium}$ CD62L$^{high}$ and CD11b$^{low}$ CD62L$^{low}$ (Fig. 4f). High expression of CD11b and shedding of CD62L indicate activated neutrophils, while high surface expression of CD62L suggests quiescent neutrophils, and loss of both surface markers is indicative of cell death. We observed a robust anticorrelation between neutrophil activation and NE2/NE4 in donors (Fig. 4g). In addition, we assessed ROS generation using CellROX and quantified the percentage of ROS-positive neutrophils for each donor (Fig. 4h). Similar to neutrophil activation, we observed an anticorrelation between ROS generation and NE2/NE4 in donors (Fig. 4i). These results suggest that individuals with an increase in pro-inflammatory neutrophils show a reduced NE2 population in the Sysmex readout. We then tracked individual neutrophil trajectories with time-lapse imaging using CellROX and Sytox. We observed that neutrophils that survived until 15 h exhibited higher ROS and extended duration with elevated ROS compared to those that died earlier (Fig. 4j–l). Together, these results demonstrate that neutrophil pro-inflammatory responses, including activation and ROS generation, delay their apoptosis, which is in turn reflected as a reduced NE2 population in Sysmex measurements.

Consistent with our finding that pro-inflammatory neutrophil responses determine the NE2/NE4 readout, our GWAS also revealed an SNP rs5743618 in the *TLR1/6/10* region associated with NE2/NE4

**Fig. 4 | NE2/NE4 measures neutrophil death and anti-correlates with neutrophil pro-inflammatory responses. a**, Flow cytometry analysis of isolated neutrophils stained with the Sysmex WDF dye (WDF-APC). NE2-like population is defined as the cells showing elevated SSC and WDF dye intensity. NE1-like population is defined as the main neutrophil population with lower SSC and WDF dye intensity. **b**, Distribution of Sytox-green intensity comparing the NE1- and NE2-like populations. **c**, Distribution of intensities of PE-conjugated Annexin V, comparing NE1- and NE2-like populations. **d**, Relationship between Sysmex NE2/NE4 readout and the percentage of Sytox-green and Annexin V-positive neutrophils from blood samples incubated at 39 °C for 17 h. Each data point represents one donor ($n = 11$). **e**, Schematics of experimental workflow for comparing neutrophil activation and ROS content at the 4.5 h time point and Sysmex readout at the 17 h time point for the same blood sample. The illustration was created with BioRender. **f**, Flow cytometry analysis of isolated neutrophils stained with Alexa 488 conjugated CD62L and Pacific blue conjugated CD11b from two representative patient samples. **g**, Relationship between the percentage of activated neutrophils at 4.5 h and the Sysmex NE2/NE4 readout at 17 h for the same donors. Each data point indicates one donor ($n = 24$). **h**, Histogram of the intensity of CellROX deep red of isolated neutrophils from two representative patient samples. **i**, Relationship between the percentage of ROS-positive neutrophils at 4.5 h and the Sysmex NE2/NE4 readout at 17 h for the same donors. Each data point represents a donor ($n = 24$). **j,k**, Time-dependent transient of CellROX deep red (ROS) and Sytox green (cell death) of neutrophils that survived till 15 h (**j**) ($n = 27$) and died before 15 h (**k**) ($n = 20$). Error bars indicate s.e.m. **l**, Time of neutrophils become ROS-positive comparing cells that survived till 15 h ($n = 27$) and died before 15 h ($n = 20$). Error bars indicate s.d. Unpaired two-sided $t$ test was used to calculate $P$ value. $P = 1.7 \times 10^{-8}$. Each data point indicates an individual neutrophil. In **d**, **g** and **i**, $R$ indicates Pearson correlation coefficient. Two-sided $P$ values are shown.

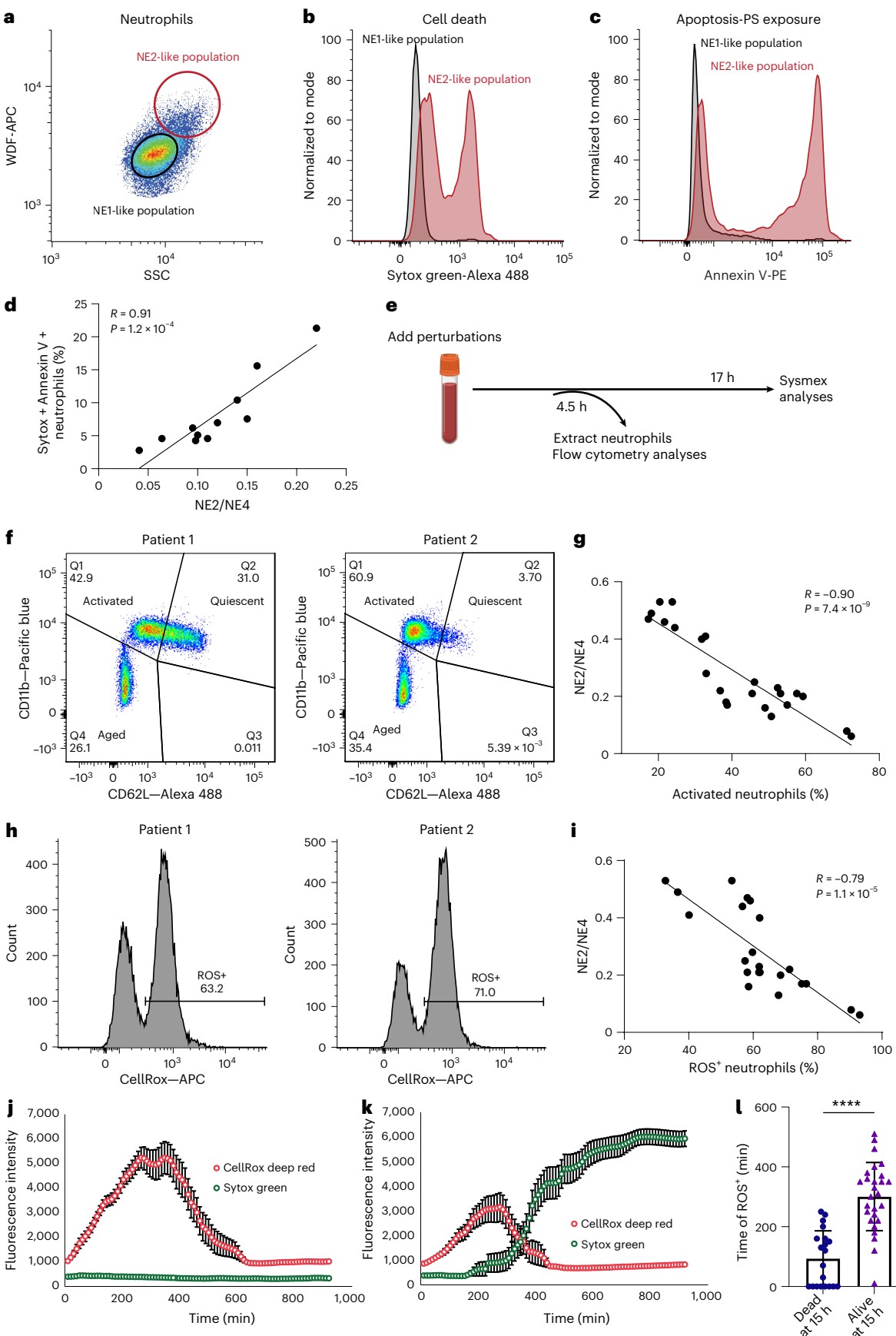

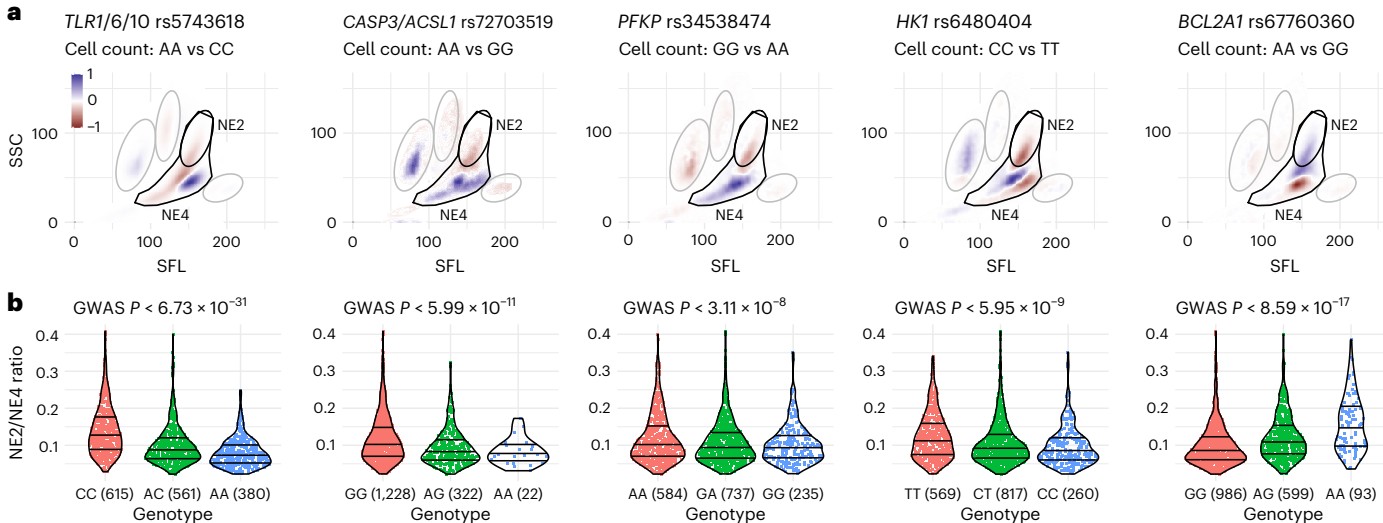

**Fig. 5 | A neutrophil population that emerges in response to inflammatory stimuli and long heat exposure is linked to multiple genetic variants.** **a**, Comparison of flow cytometry measurements in the WDF channel between homozygotes for multiple loci (*TLR1*, *CASP3/ACSL1*, *PFKP*, *HK1* and *BCL2A1*) in response to Pam3CSK4 19 h perturbation. Color gradient shows the difference in cell count distribution between homozygotes for the major and minor alleles. Cell count distributions were calculated by normalizing counts to the total cells measured for each group of homozygotes. **b**, Comparison of NE2/NE4 ratios among individuals with different genotypes at the indicated loci. Two-sided *P* values are based on *t* tests in linear regression models and are not adjusted for multiple testing. Genotype counts are shown in parentheses, with totals ranging from 1,556 to 1,678 donors for each locus. Violin plots show a smoothed distribution of all donors (points) and markings inside the violin at the 25th, 50th and 75th percentiles.

ratio (Fig. 5). This variant has been previously demonstrated to enhance TLR1 trafficking and expression on the plasma membrane and account for interindividual variability in Pam3CSK4-induced cytokine responses[30,31]. To simulate this gain of function in TLR1, we used the TLR1/2 ligand Pam3CSK4. We found a dose-dependent decrease in the NE2/NE4 profile in whole blood treated with Pam3CSK4 compared to untreated control (Extended Data Fig. 4a,b). As expected, stimulating neutrophils with Pam3CSK4 also increased neutrophil activation and ROS generation (Extended Data Fig. 4c–f). Furthermore, tracking individual neutrophil trajectories revealed that Pam3CSK4-treated cells exhibit prolonged durations of ROS elevation compared to untreated controls (Extended Data Fig. 4g). Pam3CSK4 also greatly increased neutrophils' glycolytic adenosine triphosphate (ATP) production (*P* < 0.001; Extended Data Fig. 4h), suggesting that the neutrophils undergo metabolic reprogramming after TLR stimulation, as previously observed in macrophages[32]. These results further support the role of elevated neutrophil pro-inflammatory responses underlying the decreased NE2/NE4 ratio measured with Sysmex.

## Common variants in metabolic genes regulate neutrophil activation and apoptosis

Besides *TLR1* and several genes previously reported to regulate cell death (for example, *CASP3* and *BCL2A1*)[33,34], we also identified three metabolic genes, *HK1*, *PFKP* and *ACSL1*, associated with NE2/NE4 ratio at genome-wide significance (Figs. 5 and 6a). *HK1* and *PFKP* encode hexokinase 1 and phosphofructokinase, respectively, two key enzymes regulating the rate-limiting steps in glycolysis, converting glucose into pyruvate and generating low levels of ATP[35,36] (Fig. 6a). The lead SNPs we identified for *HK1* and *PFKP* were previously associated with their increased expression (rs6480404 expression quantitative trait loci (eQTLs) for *HK1* in neutrophils: $\beta$ = 0.178, *P* = 4 × 10$^{-16}$; rs34538474 eQTL for *PFKP* in blood: $\beta$ = 0.457, *P* = 3.3 × 10$^{-310}$)[37,38]. The two SNPs were associated with a decreased NE2/NE4 ratio, suggesting reduced neutrophil apoptosis. Neutrophils are typically thought to use anaerobic glycolysis as their primary energy source. However, recent studies suggest that neutrophils use diverse metabolic pathways, including fatty acid oxidation (FAO), to provide energy for specific functions[35,36,39].

Acyl-CoA synthetase long-chain family member 1, encoded by *ACSL1*, converts fatty acid into acyl-CoA, which is then transported into mitochondria for oxidation (Fig. 6a). To investigate the effects of *HK1* and *PFKP* manipulations in neutrophils, we used a subsaturation dose of 2-deoxy-D-glucose (2-DG; 10 mM) to inhibit glycolysis and *HK1*. We used an ACSL1 inhibitor, triacsin C[40], to study ACSL1 function. We first assessed ATP production from neutrophils using the Seahorse metabolic analyzer. Consistent with the literature, we observed that unstimulated neutrophils are highly glycolytic (Fig. 6b). As expected, 2-DG decreased glycolytic ATP production (*P* = 0.03). In contrast, triacsin C ablated mitochondrial ATP production (*P* = 0.002) and increased glycolytic ATP (*P* = 4.7 × 10$^{-6}$; Fig. 6b). As ACSL1 modulates FAO, we further analyzed triacsin C's effect on FAO using exogenous palmitate as a long-chain fatty acid substrate. Compared to the DMSO control, triacsin C decreased both mitochondrial respiration and maximal respiration in response to FCCP, suggesting reduced FAO in neutrophils (Fig. 6c).

We next examined how 2-DG and triacsin C modulate neutrophil function. We found that both treatments increased the NE2/NE4 ratio in whole blood compared to controls, suggesting an increase in neutrophil death (Fig. 6d), and reduced ROS production in neutrophils (Fig. 6e,f). 2-DG also decreased neutrophil activation (Fig. 6e,g). Unexpectedly, we observed the upregulation of neutrophil activation induced by triacsin C (Fig. 6e,g). This increase is potentially caused by the metabolic shift from FAO to glycolysis in neutrophils. The bidirectional effects on neutrophil activation and ROS generation of triacsin C underlie the smaller effect on neutrophil death observed when compared to 2-DG (Fig. 6d).

Lastly, to investigate whether inhibiting *HK1*, *PFKP* or *ACSL1* promotes neutrophil apoptosis and clearance in vivo, we used a transgenic zebrafish model expressing GFP under the myeloperoxidase (*mpo*) promoter Tg (*mpo*:GFP)[41] to track neutrophil behaviors. We stimulated inflammatory responses by performing tail transection. Within 4 h post tail transection, we observed that neutrophils were recruited to the injury site, followed by resolution at around 30 h under control conditions (Fig. 6h,i). Adding a subsaturation dose of 2-DG did not alter this response (Fig. 6h,i). In contrast, under hyperglycemic conditions, at 30 h post tail transection, neutrophils continuously

accumulated at the injury site, suggesting delayed resolution of inflammation (Fig. 6h,i). Inhibiting glycolysis with a subsaturation dose of 2-DG effectively resolved prolonged neutrophil inflammation at the injury site under hyperglycemic conditions (Fig. 6h,i). In addition to pharmacological modulation, we used CRISPR–Cas9 to knockdown zebrafish orthologs *hk1*, *pfkpa/pfkpb* and *acsl1a/acsl1b*. Under control conditions, these knockdowns did not affect neutrophil recruitment or clearance (Extended Data Fig. 5). However, under hyperglycemic conditions, all three individual knockdowns promoted the resolution of neutrophil accumulation at the injury site, with *acsl1a/acsl1b* knockdown exhibiting the most pronounced effect and *hk1* knockdown showing the weakest effect (Extended Data Fig. 5).

These results suggest that *HK1*, *PFKP* and *ACSL1* interact to regulate neutrophil inflammatory responses by modulating their metabolic profiles. Pharmacological inhibition of *HK1* and *PFKP* effectively prevents sustained inflammation related to hyperglycemia and promotes neutrophil clearance. We found that SNPs leading to increased *HK1* and *PFKP* expression reduced the NE2/NE4 ratio, which is prevalent in cardiometabolic disease. Patients with these common alleles appear to exhibit delayed inflammation resolution, potentially contributing to disease pathophysiology. Thus, modulation of *HK1* and *PFKP* could serve as a mechanism-driven therapeutic strategy for such patients.

### Polygenic scores for diverse blood cell readouts predict disease outcomes

As we observed correlations between blood-response readouts and clinical traits, we sought to test whether polygenic scores (PGSs) based on blood-response summary statistics can be used to stratify patient populations and improve the predictions of clinical events. We calculated PGSs for perturbation blood responses spanning different cell types and conditions, using clumping and thresholding with fixed parameters, for participants in the Mass General Brigham (MGB) Biobank and the UK Biobank (UKBB). We first computed Cox proportional hazard models for 30 clinical outcomes, using blood-based PGSs derived from the selected 327 blood readouts, adjusting for sex and the first two genetic principal components. Then, we performed meta-analyses to identify blood traits and clinical outcomes with robust associations in both MGB Biobank and UKBB datasets.

The PGSs calculated from different blood readouts exhibited unique associations with specific diseases. We stratified participants into quartiles according to their PGS and plotted the time to first diagnosis for a subset of diseases and blood traits (Fig. 7a), which showed clear separation among different quartiles. For example, the first quartile based on PGS calculated from variability in RBC FSC under 17 h KCL perturbation (RET_KCL_17h_RBC1_SD_FSC) showed delayed onset of heart failure compared to the last three quartiles (Fig. 7a), suggesting the genetic basis underlying this blood cell trait might be used to predict risk for heart failure and explore the mechanisms leading to its

development. Because there are differences in the cohort characteristics and prevalence of outcomes between MGB Biobank and UKBB, we focused on associations that were significant in the meta-analysis of both cohorts (Fig. 7b).

We identified significant associations in both cohorts for multiple cardiometabolic conditions (Fig. 7b, Supplementary Fig. 3 and Supplementary Data 3), for example, obesity (RET_LPS_18h_RBC2_Med_SSC, $P_{adj} = 3.74 \times 10^{-6}$, MGB cases = 9,499, UKBB cases = 41,893), T2D (RET_KCl_17h_RET1_%, $P_{adj} = 1.5 \times 10^{-4}$, MGB cases = 6,226, UKBB cases = 34,941), CKD (WNR_Water_15h_WBC2_Med_FSC, $P_{adj} = 1.7 \times 10^{-5}$, MGB cases = 5,627, UKBB cases = 23,771) and heart failure (RET_KCl_17h_RBC1_SD_FSC, $P_{adj} = 6.4 \times 10^{-3}$, MGB cases = 4,421, UKBB cases = 15,811). We also observed strong associations with immune-related conditions such as type 1 diabetes (PLTF_LPS_18h_PLT_Med_SFL, $P_{adj} = 8.6 \times 10^{-5}$, MGB cases = 530, UKBB cases = 4,207), asthma (WNR_LPS_18h_WBC_Med_SSC, $P_{adj} = 8.7 \times 10^{-5}$, MGB cases = 6,176, UKBB cases = 62,009) and systemic lupus erythematosus (WDF_Alhydrogel_21h_NE2-NE4_ratio, $P_{adj} = 4.5 \times 10^{-3}$, MGB cases = 532, UKBB cases = 804). Conducting ICA based on the meta-analysis results (Fig. 7c) revealed meaningful clusters of clinical phenotypes, such as a group involving lipidemia, chronic ischemic heart disease and heart failure. These findings suggest that genetic factors influencing various blood traits can effectively stratify different disease outcomes.

### Multigenic models of *ACSL1*, *PFKP* and *HK1* predict CKD risk in patients with T2D

We further investigated blood readouts associated with variants in *ACSL1*, *PFKP* and *HK1* in detail. As demonstrated above, these metabolic genes regulate neutrophil activation and clearance, particularly in hyperglycemia. Thus, we sought to test whether PGSs calculated based on these blood cell traits predict the time to CKD onset and progression in individuals with prediabetes and diabetes (HbA1C > 5.7). We categorized CKD stages 3a, 3b, 4 and 5, based on estimated glomerular filtration rate (eGFR) thresholds (eGFR = 45–59, 30–44, 15–29 and <15 ml min$^{-1}$/1.73 m$^2$). We found that the PGSs for RBC variability in SFL under 21 h Alhydrogel, 20 h colchicine, 17 h KCL and 18 h LPS perturbations were positively associated with CKD progression, whereas the NE2/NE4 ratio under 17 h KCL, 20 h colchicine and 19 h Pam3CSK4 conditions was negatively associated with CKD development (Fig. 7d). These results suggest that PGSs based on cellular readouts can be used to identify subpopulations of disease at increased risk of discrete complications, such as accelerated progression of CKD in T2D.

## Discussion

Over 3,300 traits have been investigated using GWAS in more than 1 million participants, with current studies continuing to increase sample sizes to improve statistical power. While the techniques are robust, it remains difficult to identify underlying biological effects[6]. One major

---

**Fig. 6 | *HK1*, *PFKP* and *ACSL1* regulate neutrophils' metabolic profile and inflammatory responses. a**, Schematics showing the regulatory function of *HK1*, *PFKP* and *ACSL1* in neutrophil metabolic pathways. The lead SNPs identified are associated with upregulated expression of *HK1* and *PFKP* and unknown directionality for *ACSL1* (OpenTargets Genetics database). The illustration was created with BioRender.com. **b**, Seahorse analysis of isolated neutrophils showing ATP produced from the mitochondria and the anaerobic glycolysis pathways in conditions of water control, 2-DG, DMSO control and triacsin C. Five replicates were performed for the same donor. Error bars shown indicate s.e.m. of measurements from four donors' blood samples. **c**, Seahorse analysis long-chain FAO rate in neutrophils treated with triacsin C and DMSO control. Five replicates were performed for the same donor. Error bars shown indicate s.e.m. of measurements from four donors' blood samples. **d**, Sysmex NE2/NE4 readout of blood treated with 2-DG, compared to water as control, and triacsin C, compared to DMSO as control. Measurements were performed after 17 h incubation at 39 °C. Each data point represents a donor (*n* = 15). Error bars

indicate s.d. **e**, Flow analysis of isolated neutrophils stained with Alexa 488 conjugated CD62L, Pacific blue conjugated CD11b and CellROX. Shown are representative dot plot and histogram from one representative sample, comparing water control, 2-DG, DMSO control and triacsin C conditions. **f,g**, Percentage of ROS-positive (**f**) and activated (**g**) neutrophils at 4.5 h post-treatment of water control, 2-DG, DMSO control and triacsin C. Each data point represents a donor (*n* = 16 for control and 2-DG, *n* = 8 for DMSO control and triacsin C). Error bars indicate s.d. **h**, Visualization of neutrophils in Tg (*mpo*:GFP) zebrafish at 4 h and 30 h post tail transection, comparing control, 2-DG, hyperglycemia and 2-DG under hyperglycemia conditions. Images show four representative zebrafish. Scale bars indicate 200 μm. **i**, Quantification of GFP+ cells at the tail transection site at 4 h and 30 h under control (*n* = 5), 2-DG (*n* = 4), hyperglycemia (*n* = 7) and 2-DG under hyperglycemia (*n* = 5) conditions. Each data point indicates an individual zebrafish. Paired two-sided *t* test was used to test statistical significance shown in **d**, **f**, **g** and **i**. **\*\****P* < 0.01, **\*\*\****P* < 0.001 and **\*\*\*\****P* < 0.0001.

bottleneck is a generalizable strategy to move from a locus to a genetic target and mechanistic insights, limiting translation toward therapeutic development. We outline an approach that combines cellular phenotyping with GWAS to uncover previously latent, large effect-size genetic loci with direct implications for cell biology. Using multigenic

models based on selected cellular phenotypes, we then identified clinical phenotypes with substantially altered disease risks related to these intermediate phenotypes.

We focused on cellular responses in peripheral blood, as such samples are highly accessible and have long been used as a diagnostic tool in

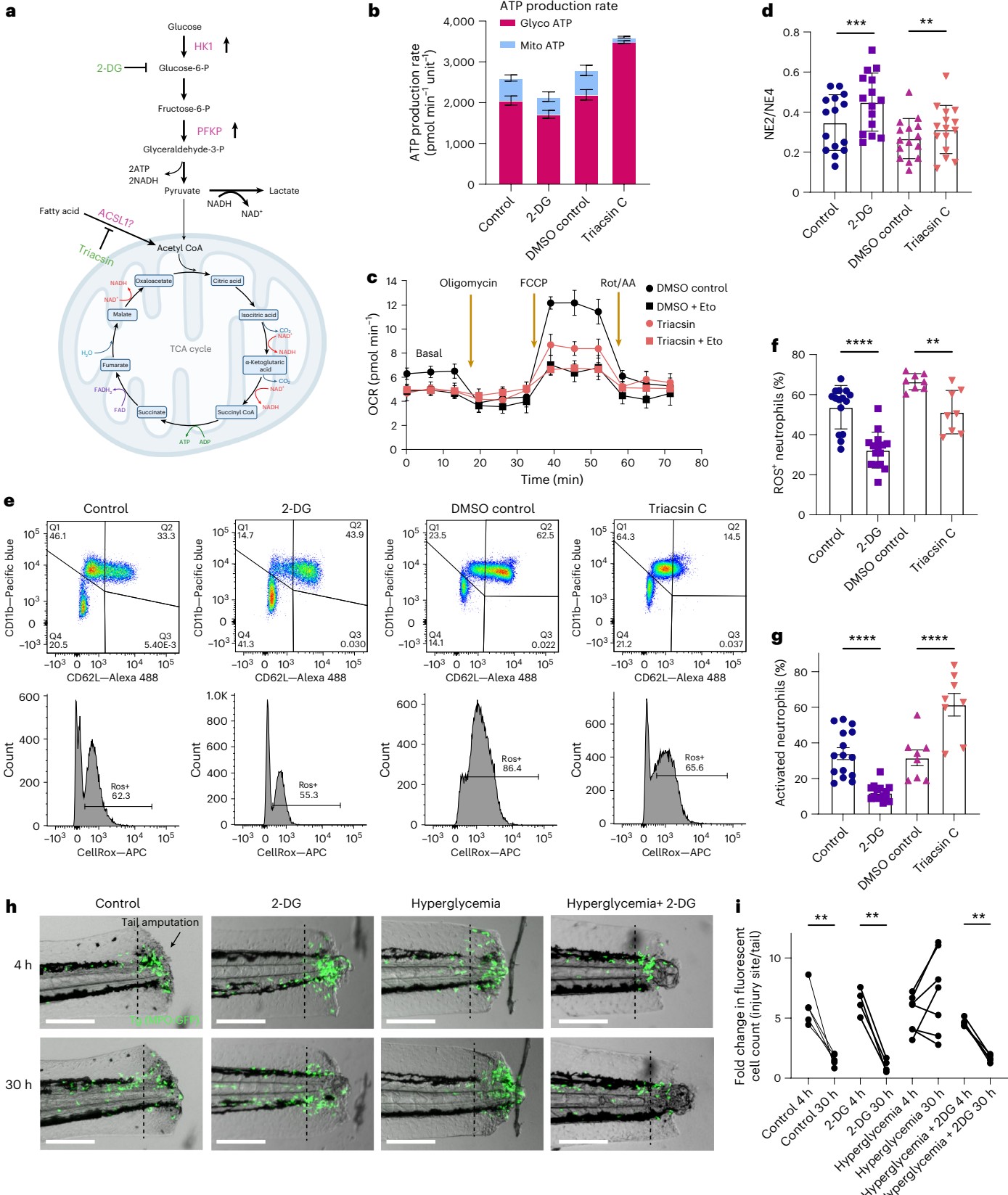

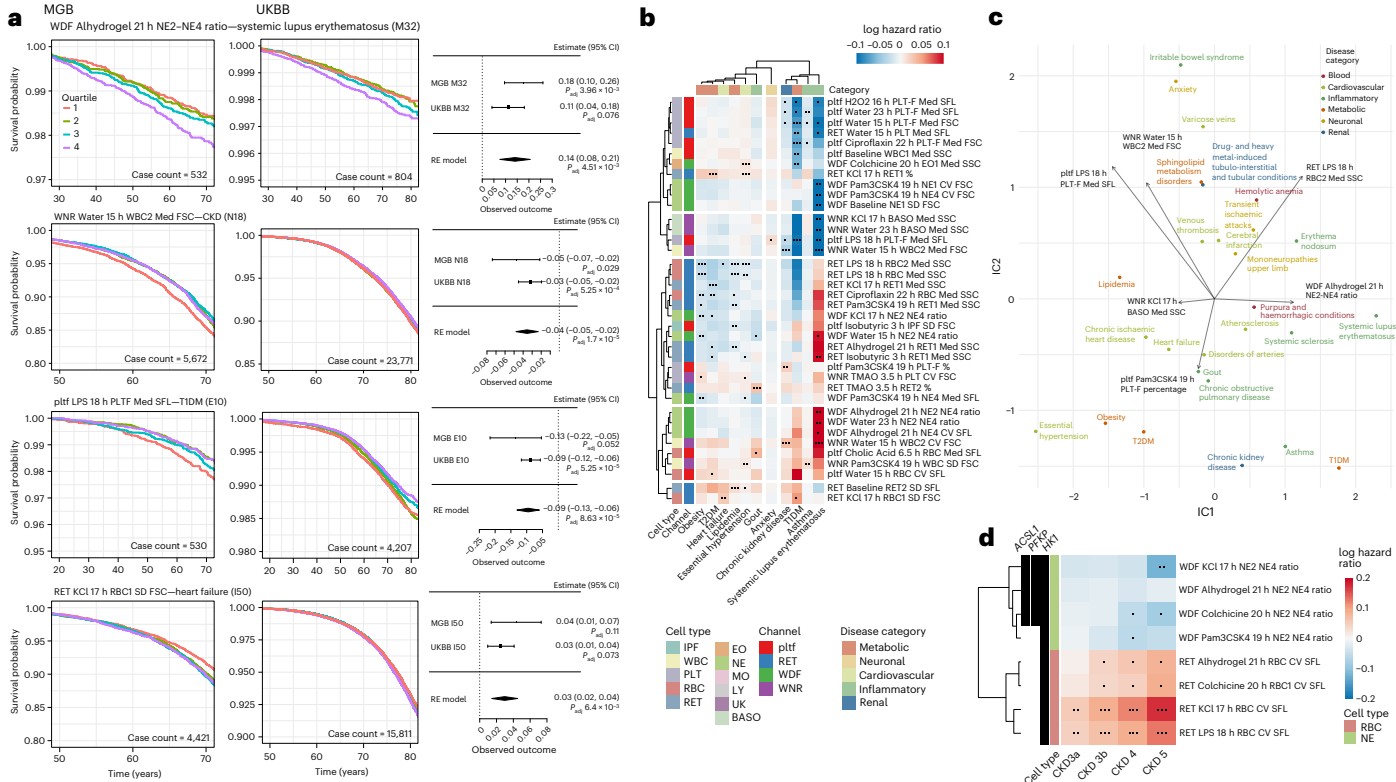

**Fig. 7 | PGSs calculated from perturbation-based blood responses are associated with differences in time to onset of diseases. a**, Survival curves and meta-analysis for diagnoses stratified by blood-response PGSs in MGB Biobank and UKBB. Time to first diagnostic code or diagnosis date in medical problem list was modeled using sex, first two genetic principal components and scaled blood-response scores in MGB and UKBB using Cox PH models with delayed entry. Meta-analysis panels show estimated log HR and 95% CI. Two-sided *P* values for MGB and UKBB were obtained from Cox PH models, and from *z* scores in a random-effect model for the meta-analyses. All *P* values are corrected for multiple testing using FDR. **b**, Hazard ratio estimates derived from time-to-event models for various clinical outcomes, using PGS of blood readouts under perturbation conditions. These estimates were based on a meta-analysis of data from the MGB Biobank and UKBB. Time to first diagnostic code or diagnosis date in medical problem list was modeled using sex, first two genetic principal components and scaled blood-response scores using Cox PH models with delayed entry. A meta-analysis was conducted to derive two-sided *P* values, using *z* scores in random-effect models that combined data from both cohorts. Points

indicate significant associations after multiple testing correction using FDR across all tested diseases and blood traits (30 clinical outcomes and 327 blood readouts) with adjusted *P* value thresholds as follows: one solid square signifies 0.05, two solid squares signify 0.01 and three solid squares signify 0.001 (see Supplementary Fig. 3 for an overview of all PGS-disease associations). **c**, ICA of the association score matrix between blood readout PGSs and clinical endpoints. A subset of diagnoses and lab values projected onto the first two components together with mixing matrix loadings of selected blood readouts is shown. **d**, Hazard ratio estimates for the progression to different CKD stages in individuals with prediabetes and diabetes using PGS of blood traits that had significant associations with *ACSL1*, *PFKP* or *HK1* in the MGB Biobank. Cox PH models were applied to analyze time until the initial diagnosis of each CKD stage, using two-sided tests for statistical evaluation. Points indicate significant associations after multiple testing correction using FDR with adjusted *P* value thresholds as follows: one solid square signifies 0.05, two solid squares signify 0.01 and three solid squares signify 0.001.

---

clinical settings, and technologies are broadly available for subsequent scaling of any useful findings. In addition to clinically available assays of cross-sectional cellular counts, we assessed blood cell properties under 36 perturbation conditions, aiming to elicit phenotypes that are latent at baseline, and thus likely to be previously unmeasured. We chose this approach to favor the identification of new disease-related endophenotypes, from which we could select those associated with large effect size common alleles that might represent rigorous drug targets. We expanded the phenotypic space from 29 blood parameters used in previous studies to over 4,000 cell readouts. We were able to identify alleles associated with key cellular processes, such as neutrophil activation and apoptosis, which have roles in common complex diseases beyond hematopoietic disorders. Evoked cellular response traits in peripheral blood offer a complementary approach to existing phenotyping with the potential to identify genes and pathways with translational and clinical relevance.

To validate that risk genes identified using our framework are linked to disease-relevant biology, we conducted functional studies of genes associated with the evoked NE2 population. Although the Sysmex

measurements are not tailored to characterize neutrophil function, we found that WDF (a nucleic acid dye) used to distinguish blood cell lineages is reflective of neutrophil apoptosis. We further elucidated that the delay in neutrophil apoptosis was due to a neutrophil pro-inflammatory response. The perturbation-based assays we developed enabled the efficient identification and experimental validation of genes (*HK1*, *PFKP* and *ACSL1*) involved in metabolic pathways affecting neutrophil ROS generation and lifespan, revealing cell metabolism as a potential therapeutic target for inflammation in various cardiometabolic diseases.

Our approach reveals common genetic variants with large effect sizes. Notably, several genes we identified have been previously demonstrated to underlie specific Mendelian diseases. For example, we identified common coding variants in *TUBB1* that affect platelet traits, while rare variants in *TUBB1* were previously linked to inherited thrombocytopenia[42,43]. *BMPR2*, which is linked to hereditary pulmonary arterial hypertension (PAH)[44], was associated with monocyte responses in this study. As monocytes and macrophage abnormalities have been implicated in the pathophysiology of PAH[45], this finding suggests a monocytic contribution to the vascular inflammation observed in

*BMPR2*-linked PAH but also offers a window into potential somatic contributions to other forms of PAH. These examples support the utility of latent phenotypes to define cellular mechanisms that can bridge common genetic variation and complex diseases.

PGSs calculated from a subset of blood cell traits associated with metabolic genes showed utility in risk prediction for renal complications of diabetes. Emerging evidence supports the involvement of innate immunity in CKD initiation and progression in diabetes, but studies have typically focused on macrophages[46]. Our results reveal a role for genetically determined variation in the genesis of pro-inflammatory neutrophils in CKD development in diabetic patients. The PGS models based on blood readouts were able to stratify patients with distinct risks for developing various cardiometabolic, vascular and inflammatory diseases, revealing subgroups that might benefit from therapeutics targeting related biological pathways.

Our study has several limitations. Firstly, we used a conventional significance threshold of $P < 5 \times 10^{-8}$ for genetic association without adjusting for the number of phenotypes tested, which may result in false positives. We estimated that approximately 350 traits were independent among the phenotypes tested. To reduce the false discovery rate (FDR), we reported significant associations only when at least two independent traits were linked to the clumped region. In practice, the evoked cellular traits and their genetics are efficiently validated in scalable in vivo models. Secondly, we had varying sample sizes across different perturbations, which could reduce statistical power for conditions with fewer samples, potentially resulting in false negatives. Furthermore, while our phenotypic associations are derived from multiple ancestry groups, the genetic associations are based on individuals of European ancestry due to limited representation of other ancestry groups in our cohort. We performed GWAS analyses for a subset of blood cellular traits across multiple ancestry groups, which revealed consistent trends in effect directions, albeit with notable disparities for several lead SNPs (Extended Data Fig. 6). Future investigations are needed to unravel the *trans*-ancestry genetic basis governing evoked blood responses. Lastly, for PGSs related to clinical traits based on EHR, we employed Cox proportional hazard models (time-to-event analyses). However, EHR data inherently present limitations, because they do not capture the entire medical history and there can be misalignment of the age of disease onset versus diagnosis. To address these issues, we used Cox models with delayed entry to handle incomplete observations. Nevertheless, the time of disease onset could be misrepresented due to the inherent constraints of EHR data.

In summary, we performed perturbational blood cell phenotyping using a widely available cytometry device that is primarily designed for robust whole-blood cell counts. This framework incorporating human genetic data, primary cellular phenotyping and deep clinical traits enables the iteration of genetic risk locus discovery, systematic target validation and subsequent drug discovery. Implementing such a method in routine clinical settings will facilitate the development of refined clinical trajectories and identification of large effect size common variants contributing to human disease and clinical outcomes.

## Online content

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

## Methods

### Human study participants

Study participants were recruited in accordance with IRB 2019P003155 from multiple phlebotomy clinics in the MGB hospital system. Sample sizes of measured blood profiles and genotyped subjects per perturbation condition are listed in Supplementary Table 1. Demographic information such as age and sex are provided in Supplementary Table 3. Written informed consent was obtained from all individuals. The MGB Institutional Review Board approved the analyses of the UKBB (application 55482).

### Zebrafish

All zebrafish studies were carried out under the protocols approved by the Brigham and Women's Hospital Standing Committee on Animals.

### Reagents

Details of reagents used in this study are included in Supplementary Table 5.

### Whole-blood perturbation screening

Physiologically relevant doses and time points were determined for each perturbation to elicit reproducible effects on blood analyzed on a Sysmex XN-1000 hematology analyzer (see Supplementary Table 1 for perturbation condition descriptions including dose and exposure times, and Supplementary Table 5 for the details of chemical agents). Compounds dissolved in DMSO or chloroform were prepared such that the percent by volume of solvent is <0.5%. Each condition was assigned a three-digit identifier (for example, −007) that was paired with a patient ID for each treated sample (for example, AA-00100-007). This standardized label scheme allowed for the preparation of barcoded sample tubes and batch-wise automated measurements using the hematology analyzers. Sysmex XN-1000 was calibrated each day using Sysmex XN Check levels 1–3. New QC lots were acquired every 28 d as recommended by the manufacturer's guidelines.

Up to 40 individuals per day were recruited from multiple phlebotomy clinics and donated up to 50 ml of blood in addition to their clinical blood draw. Whole blood was collected in 8.5 ml ACD tubes (BD 364606). Barcoded sample tubes with patient and perturbation identifiers were aligned and prepared batch-wise, by aliquoting 700 µl of whole blood into a grid of 5 ml round bottom tubes. All perturbation compounds were added to blood at specific time points and transferred to incubator shakers (39 °C, 200 RPM). After incubation, tubes were placed in automated sampling racks and profiled using the Sysmex XN-1000. Both Sysmex-derived blood parameters (for example, CBC) and raw cytometry data were exported as .csv and .fcs files.

### Genotyping, quality control and imputation in screening cohort

Before aliquoting patient blood samples, a portion of freshly drawn blood was set aside for whole-blood DNA extraction. DNA was extracted from 3 ml of whole blood using Qiagen Puregene Blood Core Kit C (158389). DNA was quantified and checked for quality using NanoDrop One and Qubit, diluted to 75 ng µl$^{-1}$ and stored at −80 °C. Samples were aliquoted into 96-well barcoded plates and quantified using Cytation Take3 Trio before genotyping. Internal genotyping for quality control was performed using Advanta Sample ID Genotyping Panel (Fluidigm, 101-7773). Aliquots were shipped to Northwell Health Genomics Alliance and the University of Miami Genotyping Core in 96-well barcoded plates with one empty well for controls. Samples were quantified using Nanodrop and Qubit to identify plates with high numbers of low-concentration samples, which could be replaced before genotyping. Genotypes were called from genomic DNA in batches of approximately 500 samples using the Illumina GSAv3 Beadchip and Illumina Genome Studio.

Computational analyses used Python 3.9 and R 4.2. Genotype data were processed using PLINK1.9 and PLINK2 (ref. 47). Samples were excluded from participants who had high missingness of variants (>10%), had sex mismatches from genotyped data or had withdrawn from the study. In addition, for samples failing Advanta fingerprinting (concordance of at least 0.75 in at least 20 SNPs), genotyping was repeated, or the samples were removed. Variants with high missingness across individuals (>10%) or deviations from Hardy–Weinberg equilibrium at $P < 1 \times 10^{-50}$ were filtered. Structural or multi-allelic variants were removed. A local instance of Michigan Imputation Server v1.5.7 (ref. 48) with Eagle2 and Minimac4 was used to impute genotypes with the 1000G Phase3 v5 reference panel. After imputation, variants with minor allele frequency of <0.0001 were removed. The first ten principal components were estimated using PLINK2. Relatedness was estimated using PLINK2 with the KING-robust kinship estimator[49] and five individuals with a kinship greater than 0.177 (first-degree relations or closer) were removed. In total, after these exclusions, genotype data were available for 2,685 individuals on >3.5 million imputed variants. Based on self-reported ancestry at study entry, our cohort consisted primarily of individuals with European ancestry, preventing robust multi-ancestry analyses due to low numbers of individuals in other ancestry groups. Therefore, we calculated and reported genetic associations for the subset of participants with self-reported European ancestry only (discovery cohort). For cross-ancestry validation of the lead variants, we used the following self-reported ancestry groups: AFR, ASIAN and OTHER (including Other, Pacific Islander and Native American) for separate GWAS analyses. Genotyped individuals in the self-reported HISPANIC group were not included in the cross-ancestry analyses due to insufficient numbers.

### Genotyping, quality control and imputation in MGB Biobank cohort

MGB Biobank samples were genotyped in batches using three related Illumina arrays (MEGA, MEGA Ex and MEG), as well as the Illumina GSAv3 array. Imputation was performed using the Michigan Imputation Server with the 1000G Phase3 v5 reference panel for each batch. We merged batches using the intersection of variants present in all batches and applied the same QC filtering as above. In short, individuals with high missingness (>10%) or sex mismatches were removed. Variants with high missingness across individuals (>10%) or deviations from Hardy–Weinberg equilibrium at $P < 1 \times 10^{-50}$ were filtered. Structural or multi-allelic variants were removed. Principal component analysis (PCA) was calculated using PLINK2, and individuals with a kinship greater than 0.177 as well as individuals with non-European ancestry (distance greater than 3× radius of 1000G EUR reference samples in joint PCA) were removed using plinkQC[50]. Individuals who were part of the screening cohort were removed from the MGB Biobank cohort. In total, after these exclusions, genotype data were available for 44,705 participants on >6.7 million imputed variants. For PGS applications, we further filtered variants to have a minimum minor allele count of 100 and missingness <2%, leaving 1.8 million variants.

### Genotyping, quality control and imputation in UKBB cohort

The UKBB samples were genotyped on two Affymetrix arrays, UK BiLEVE and UKBB Axiom. The genotyping data underwent stringent quality control procedures described elsewhere[51], including exclusion of individuals based on missingness, heterozygosity, sex mismatch, relatedness and non-British ancestry. Imputation was carried out using a two-step prephasing/imputation process using SHAPEIT and IMPUTE2 software, using the Haplotype Reference Consortium and UK10K haplotype resources. Post-imputation quality control included the removal of variants with minor allele frequency <1%, minor allele count >100, variants with an imputation quality score (Minimac $r^2$) < 0.4 and those not in Hardy–Weinberg equilibrium ($P < 1 \times 10^{-15}$). We used the White ethnic background cohort based on the self-reported UKBB

data field f21000. After these quality control steps, data for approximately 424,000 participants with clinical outcomes were available. PCA was performed on the non-imputed genotype data of the same individuals using PLINK2.

## Phenotype measurements and quality control

We measured a total of 278 blood-based cellular phenotypes using a blood flow cytometer (Sysmex XN-1000) under 37 different conditions. The blood cell parameters can be categorized into indices related to membrane/intracellular structure measured using SSC, nucleic acid and membrane lipid content measured using SFL, and cell shape/volume measured using FSC, as well as parameters such as cell counts and percentages within defined regions (gates). For each parameter, we calculated robust estimators such as median, robust s.d. and robust coefficient of variation using FlowJo v10.8. Gates were empirically defined based on densities of measured cells under baseline and perturbation conditions and included additional regions for subpopulations that were typically not observed under baseline conditions. We defined a total of 15 WBC-related gates, 7 RBC gates, 4 platelet-related gates and 4 gates for debris or unknown cell types. All samples were measured within 36 h of blood draw, with baseline measurements occurring within 3 h for 80% and 7 h for 95% of samples.

We performed thorough quality control to identify sources of technical variation as well as biological covariates. For this, we assessed the effect of the time between blood draw and flow cytometry measurement, drift over the course of the study (study month) and biological covariates such as age, sex and race (Supplementary Fig. 4). We removed outlier samples where a single phenotype was outside of four median absolute deviations from the median measurement of all samples under the same conditions. We also computed a two-dimensional ICA projection for all blood measurements from a single fluorophore under a single perturbation condition and removed samples that were further than 2.5 median absolute deviations from the median sample. Finally, we quantile-transformed the phenotypic measurements. The final numbers of blood measurements as well as genotyped individuals passing QC across conditions are shown in Supplementary Table 3.

## Estimation of the number of independent traits

During the study, multiple batches of perturbations were administered across different time periods, each involving mostly nonoverlapping groups of individuals. Due to the distinct cohorts and perturbation conditions across batches, the data consisted of several mostly complete blocks of measurements (apart from missing values in individual measurements). We approached each of these blocks separately to estimate the effective number of independent traits. To estimate the effective number of independent traits, we used quantile transformation followed by PCA on each of these blocks of blood readouts separately. We used the R package 'PCAtools' v2.12.0 to determine the count of PCA components that cumulatively explained 90% of the variance in the data for each block. This number varied from 243 to 349 across the blocks. However, the blocks also shared a subset of perturbation conditions, and we observed recurrent genetic associations under different perturbations, suggesting an overlap of underlying structure. Based on these analyses, we estimate the presence of over 350 independent traits (Supplementary Fig. 5).

## Flow cytometry

Flow cytometry analyses were performed on neutrophils isolated from patients' whole-blood samples, using the EasySep Direct Human Neutrophil Isolation Kit (STEMCELL, 19666). After isolation, neutrophils were resuspended in Tyrode's solution as described previously. To characterize the NE2-like cell population using flow cytometry, neutrophils were isolated from whole-blood samples that were incubated at 37 °C for 17 h and then labeled with apoptosis indicators, Sytox green (Thermo Fisher Scientific, S7020) and R-PE conjugated Annexin V

(Thermo Fisher Scientific). The labeled neutrophils were then subjected to permeabilization using Sysmex WDF Lysercell (Sysmex) and staining with Fluorocell WDF dye (Sysmex). The samples were analyzed for 5 min after the addition of Fluorocell WDF dye.

To characterize neutrophil activation and ROS, isolated neutrophils were labeled with Pacific Blue anti-human CD11b antibody (BioLegend, Clone ICRF44, 1:100 dilution) and Alexa Fluor 488 anti-human CD62L antibody (BioLegend, Clone DREG-56, 1:100 dilution). Cells were then subsequentially labeled with CellROX Deep Red Reagent (Thermo Fisher Scientific, C10422) at 37 °C for 30 min. Cells were washed and resuspended in staining buffer before flow cytometry analyses.

## Seahorse metabolic analysis

For the real-time ATP rate assay, a DMEM assay medium containing 10 mM glucose, 1 mM pyruvate and 2 mM glutamine was used. Extracellular acidification rate and oxygen consumption rate were measured from neutrophils isolated from patients' whole blood pretreated with or without 2-DG (10 mM) or triacsin C (5 µg ml$^{-1}$), using a Seahorse XFe96 analyzer. Neutrophils were resuspended in DMEM medium and seeded ($1 \times 10^6$ per well) in a Seahorse 96-well plate coated with CellTak (Corning, 354240) for 20 min. Cell attachment was visually confirmed before the assay. The assay was performed according to manufacturer instructions. Here 1.5 µM oligomycin, 1 µM FCCP and 0.5 µM rotenone/antimycin A were used.

For the long-chain fatty acid stress test, neutrophils isolated from untreated whole blood were first resuspended and incubated for 2 h at 37 °C in a substrate-limited medium containing 0.5 mM glucose, 1 mM glutamine, 0.5 mM L-Carnitine, and 1% FBS. Cells were then pelleted and resuspended in an assay medium containing 2 mM glucose and 0.5 mM L-Carnitine. Cells were seeded ($1 \times 10^6$ per well) in a Seahorse 96-well plate coated with CellTak (Corning, 354240) for 20 min. After visually confirming cell attachment, cells were treated with triacsin C (5 µg ml$^{-1}$) or DMSO control for 30 min. Palmitate-BSA FAO substrate was added before the assay. The assay was performed according to the manufacturer's instructions. Also, 4 µM etomoxir, 1.5 µM oligomycin, 1 µM FCCP and 0.5 µM rotenone/antimycin A were used. Normalization for both assays was performed based on direct cell counting.

## Zebrafish tail transection and hyperglycemia induction

Zebrafish larvae at 54 h postfertilization were anesthetized by immersion in E3 water with 4.2% tricaine. Tail transections were performed with a sterile scalpel at the distal end of the notochord. Brightfield and fluorescence images were acquired with a Cytation 5 at 4 h, and 24 h or 30 h post-transection at 28 °C. A neutrophil count within the tail region was performed using ImageJ. We induced hyperglycemia in zebrafish larvae by ablating $\beta$-cells as previously described[12]. Briefly, 48 hpf embryos were treated with 500 µM alloxan for 30 min, followed by incubation in E3 water containing 30 mM glucose.

## Zebrafish genetic knockdowns

The *hk1*, *pfkpa/pfkpb* and *acsl1a/acsl1b* knockdown zebrafish lines were generated using CRISPR–Cas9. Two-part guide RNAs were used to knockdown each gene. The guide RNAs were designed using CHOPCHOP[52], targeting the sequences shown in Supplementary Table 6. CRISPR RNAs (crRNAs) were synthesized (Integrated DNA Technology) and then annealed with *trans*-activating crRNA (tracrRNA) and incubated with Alt-R Cas9 Nuclease to form the ribonucleoprotein complex. Here 1.5 nl of the complex was injected into Tg (*mpo*:GFP) embryos at the one-cell stage.

## Genome-wide association tests and model selection

After genetic and phenotypic QC, blood phenotypes were retained for 4,723 individuals and genotypes for 2,685 individuals. We excluded debris, ghost and NRBC cell-type gates from genetic association tests

because they yielded non-normally distributed phenotypes after quantile transformation. We performed an univariable GWAS for each of the remaining 278 traits under 37 different conditions. Specifically, we used PLINK2 to compute association statistics for a linear regression of phenotype on the allele dose for >3.5 million imputed variants with minor allele frequency >0.05, minor allele count >10, covariate variance standardization and the covariates age, sex, time from blood draw to analysis, month of study, genotyping chip and batch and the first ten genotype principal components.

We used $P < 5 \times 10^{-8}$ as a significance threshold for each phenotype and did not correct for multiple testing at the level of association $P$ values. Many of our measured phenotypes were correlated across similar gate/cell types (for example, subpopulations of neutrophils), phenotypic dimensions (for example, SSC and FSC) or conditions (for example, TLR ligands Pam3CSK4 and LPS). Given the large number of tests and limited number of study participants, we sought to identify a concise set of variants that are associated with the strongest observed cellular responses. For this, we clumped all significant variants using PLINK1.9 with LD $r^2 > 0.50$, physical distance <250 kb between clumped variants and at least two independent hits from different traits for each clumped region. We used the variant with the smallest association $P$ value across all measured traits for a given region as the lead variant. The following command was used for clumping and gene range annotations: plink --clump-range glist-hg19 --clump-p1 0.00000005 --clump-p2 0.00000005 --clump-r2 0.50 --clump-kb 250 --clump-replicate --clump {trait_files}. This command also annotated associated regions using gene range lists provided by PLINK2 (https://www.cog-genomics.org/static/bin/plink/glist-hg19). If multiple genes were present for a given location, we used the locus-to-gene model from OpenTargets Genetics to identify likely candidates[53]. We prioritized candidate genes in the following order: coding variants, variants in introns and distance to transcription start sites. If there was no clear evidence for a subset of candidates, we reported the full list from the PLINK gene annotation step. We also annotated each region with associations previously reported for blood cell traits based on the supplementary material of ref. 17.

## Association with clinical phenotypes in the screening cohort

We defined 30 binary clinical phenotypes using ICD10 diagnostic codes (Supplementary Table 4). We also collected 20 quantitative measurements available across our entire cohort such as the comprehensive metabolic panel, lipid panel and structured electrocardiographic data. We fitted logistic or linear models associating binary and continuous traits with 327 blood phenotypes (top three traits with the lowest GWAS $P$ value were selected for each unique locus). Blood readouts were quantile transformed and models included the covariates age, sex, race and time from blood draw to measurement. For categorical outcomes, we used the 'glm' function in 'statsmodels' 0.13.2 with the formula 'diagnosis~blood_readout+age+race+sex+draw_time' and binomial family linkage. For continuous outcomes, we used the 'ols' function in 'statsmodels' with the same formula. Models for categorical and continuous outcomes were tested using $z$ test and $t$ test, respectively. Subsequently, to control the FDR in the presence of multiple comparisons, we computed $q$ values using the 'qvalue' package v2.4.2 in R. The $q$ values provided an estimate of the minimum FDR at which each test may be considered significant. A listing of clinical associations including covariates, case counts, $\beta$ coefficients and adjusted $P$ values is provided in Supplementary Data 2.

## PGSs and disease associations in the MGB and UKBB cohorts

For 327 traits with significant genetic associations, we used summary statistics from the screening cohort to calculate PGSeters to calculate PGSs. Specifically, we used the command plink --clump-p1 0.5 --clump-r2 0.5 --clump-kb 100 for clumping and a $P$ value threshold of 0.1 for the scoring step.

Our survival analyses model the time to first observed diagnosis after birth, considering the age at the first available diagnosis for any diagnostic code as the start of the observation or 'delayed entry' into the model. We use the framework of counting processes to account for this delayed entry, and the corresponding survival models are fit using Cox's proportional hazards regression. Counting process models allow us to consider each individual's date of birth as the starting point while acknowledging that our observation period for each individual only starts at their first hospital or outpatient visit that is documented in the EHR.

There are two settings in which we define events as having occurred between birth and the beginning of the observation period. Cases where previous medical history (only available in MGB cohort) contains the diagnoses of interest, but without a specific diagnosis date, were treated as the disease onset occurring at some unknown time in the interval between birth and start of observation period (for example, before the first hospital encounter). In addition, if the time between the start of the observation period and the event date in the EHR system is less than 1 year, we assume that the true event date most likely occurred between birth and the first visit in the healthcare network and was only reported in the EHR with delay. In these cases, we consider it an 'instant event' and encode it as having occurred in the interval between birth and start of the observation period.

We used the same disease definitions as above (Supplementary Table 4) to define case status, as well as the age at first diagnostic code or first mention in the medical problem list as event date. We calculated Cox proportional hazard models for the time to onset of 30 clinical outcomes with the variables sex, first two genetic principal components and PGS for 327 blood traits using the R package ('survival' 3.5-3), which provides support for survival analyses based on counting processes including delayed entry. For a visual comparison of study participants, we also stratified individuals into PGS quartiles and plotted Kaplan–Meier curves.

## Meta-analyses of MGB and UKBB disease associations

To integrate the results from the MGB Biobank and the UKBB, we conducted a meta-analysis on each blood PGS−clinical endpoint model using the 'rma' function from the 'metafor' package in R. We fitted a random-effect model using the restricted maximum likelihood method, which allows for the potential heterogeneity of effects across datasets. We used the estimated log hazard ratios and their standard errors from each dataset as inputs to this model and visualized the results with forest plots. To control the FDR in the presence of multiple comparisons, we computed $q$ values using the 'qvalue' package v2.4.2 in R. Listings of PGS associations at the meta-analysis stage as well as in MGB and UKBB are provided in Supplementary Data 3–5.

## ICA of blood traits and clinical endpoints

To visualize the multivariate structure between blood traits and clinical endpoints, we used an ICA of the association $t$ scores calculated for blood readouts in the screening cohort, as well as blood-trait PGS association $t$ scores calculated in the meta-analysis step. The matrix of $t$ scores thus represented the pattern of association between all pairs of blood traits and clinical endpoints across our data. We conducted ICA using the 'fastICA' R package v1.2−3. This computational method separates a multivariate signal into additive subcomponents that are maximally independent. Applying the ICA to our matrix resulted in the following two outputs: a set of independent components and a mixing matrix. The independent components represented dimensions of variation within the data, while the mixing matrix showed how each original variable (blood readout or blood-based PGS) contributed to these dimensions. To visualize our results, we plotted the first two independent components, which gave us a projection of clinical endpoints into a two-dimensional space. We also used the weights from

the mixing matrix to indicate the direction of association for a subset of blood traits within this space.

## Additional statistical analysis

We first assessed the normality of the data with the Kolmogorov–Smirnov test. If the distribution was normal, for comparisons between the two groups, we used an unpaired two-tailed Student's $t$ test. For comparisons between treatments for the same donor, we performed paired two-tailed Student's $t$ tests. When the data were not normally distributed, we used the nonparametric Mann–Whitney test for comparison between two groups and the Wilcoxon matched-pair signed-rank test for comparison between different treatments for the same donors. To assess statistical significance in difference across more than two groups, we used an ordinary one-way analysis of variance test followed by Dunnett's multiple comparison test.

## Reporting summary

Further information on research design is available in the Nature Portfolio Reporting Summary linked to this article.

## Data availability

Individual-level data are subject to restrictions imposed by patient consent and local ethics review boards. GWAS summary statistics have been deposited in the GWAS catalog database (GCST90257015-GCST90257105). PGSs as used for the UKBB analyses have been deposited in Figshare (https://doi.org/10.6084/m9.figshare.24354235). Clumped significant variants are listed in Supplementary Data 1. Clinical outcomes and quantitative lab measurements associated with blood readouts with $P_{adj} < 0.1$ are listed in Supplementary Data 2. Clinical outcomes associated with polygenic models derived from blood readouts with $P_{adj} < 0.1$ are listed in Supplementary Data 3 for the meta-analyses, and Supplementary Data 4 and 5 for the MGB and UKBB cohorts, respectively. Other datasets generated or analyzed during the current study can be made available upon reasonable request to the corresponding authors.

## Code availability

The custom code used in this study is available at https://doi.org/10.5281/zenodo.10041992 (ref. 54). For proprietary or commercial software/tools used in this study, please refer to the materials and methods section for details on how to access them or contact the corresponding author for more information.

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

## Acknowledgements

This work was supported by One Brave Idea, cofounded by the American Heart Association and Verily with significant support from AstraZeneca and pillar support from Quest Diagnostics (to C.A.M. and R.C.D.). M.H., W.Z. and S.G. are supported by the Tobia and Morton Mower Science Innovation Fund Fellowship.

## Author contributions

M.H. and W.Z. designed and performed experiments and data analyses, and drafted the manuscript. S.S.E., P.C.T. and H.Z. designed and performed experiments and data analyses. C.N.W., D.D.K., L.L.X., C.N., Z.S., J.C., C.G.E., M.N.H., A.S.T., T.M., S.G., J.G.T., B.W. and S.V. performed experiments and provided technical assistance. E.W., C.S., J.B.N., D.N.N., G.M.L., H.C.F., C.J.P., M.C., S.S. and C.R. performed and coordinated the recruitment of study participants. R.C.D., M.H., C.A.M., S.V., and W.Z. contributed to the study conceptualization and design, and edited the manuscript. All authors read and approved the final version of the manuscript.

## Competing interests

R.C.D. was supported by grants from the National Institutes of Health and the American Heart Association (One Brave Idea, Apple Heart and Movement Study) and is a cofounder of Atman Health. C.A.M. is supported by grants from the National Institutes of Health and the American Heart Association (One Brave Idea, Apple Heart and Movement Study); is a consultant for Bayer, Biosymetrics, Clarify Health, Dewpoint Therapeutics, Dinaqor, Dr. Evidence, Foresite Labs, Insmed, Pfizer and Purpose Life Sciences; and is a cofounder of Atman Health. R.C.D., M.H., C.A.M., S.V. and W.Z. are co-inventors on patents related to this work, and C.A.M., R.C.D., M.H. and W.Z. hold equity in Tanaist. All other authors report no competing interests.

## Additional information

**Extended data** is available for this paper at https://doi.org/10.1038/s41588-023-01600-x.

**Correspondence and requests for materials** should be addressed to Max Homilius, Wandi Zhu, Calum A. MacRae or Rahul C. Deo.

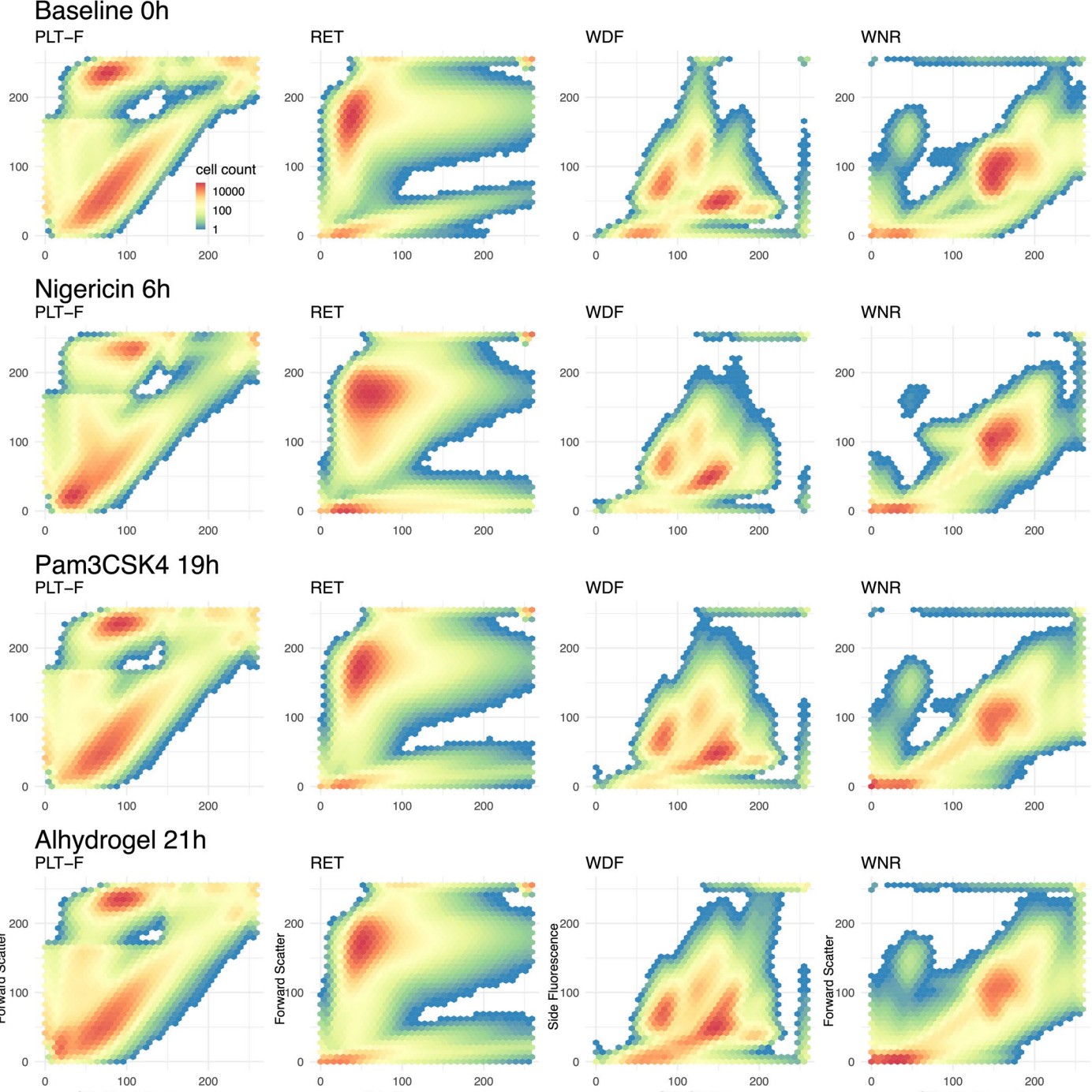

**Extended Data Fig. 1 | Blood cell distributions under baseline and perturbation conditions.** Distribution of blood cytometry readouts under baseline and three perturbation conditions. Aggregate counts for each bin were calculated across all samples for each condition and normalized to the number of cells measured in each channel. Each channel records three dimensions (forward scatter, side scatter and side fluorescence). Plots show the two dimensions used for gating cell types in each channel.

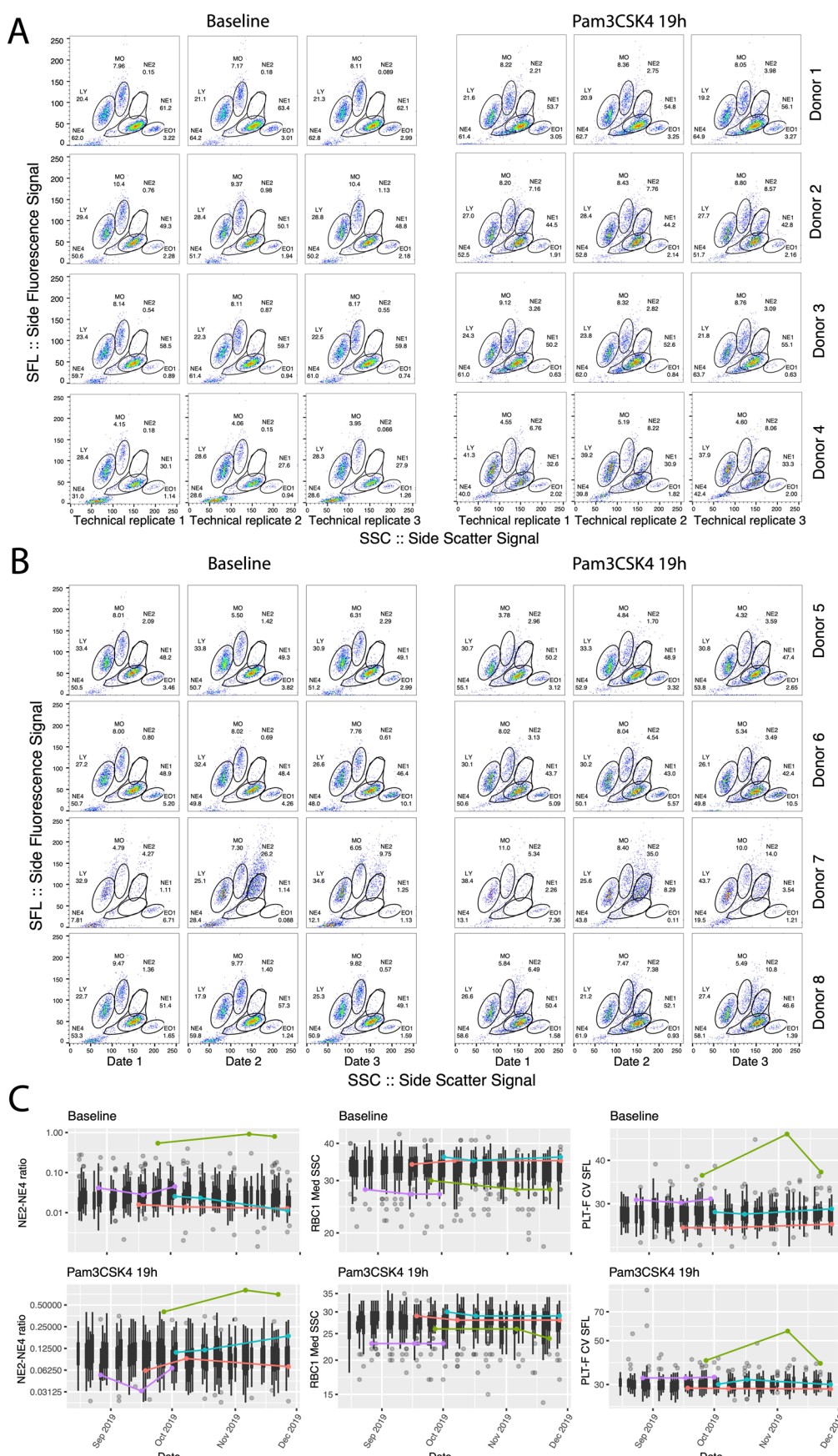

**Extended Data Fig. 2 | See next page for caption.**

**Extended Data Fig. 2 | Blood cell distribution and gates in WDF Channel for technical replicates from the same blood draw and measurements from the same individuals over time. a**, Examples of blood cell distributions in the WDF channel for three technical replicates of four randomly selected donors under baseline and Pam3CSK4 19 h conditions. Replicates were performed on samples collected from the same blood draw. **b**, Examples of blood cell distributions in the WDF channel for three longitudinal replicates of four randomly selected donors under baseline and Pam3CSK4 19 h conditions. Replicates were performed on the same donor from samples collected at different time points that are months apart. **c**, Examples of three blood traits calculated from our flow cytometry gates (NE2/NE4 ratio, RBC1 Med SSC, and PLT-F CV SFL) under baseline and Pam3CSK4 19 h conditions for data collected over the course of four months. The black dots shown indicate all study participants. The time-dependent trajectories of four donors (same individuals as shown in **b**) were plotted in colors. Boxplots for daily measurements represent the interquartile range (IQR) between the first and third quartiles as the box, the median as the line inside the box, and the whiskers extend from the box to the largest and smallest values within 1.5× IQR, with any points outside of this range shown as individual outliers.

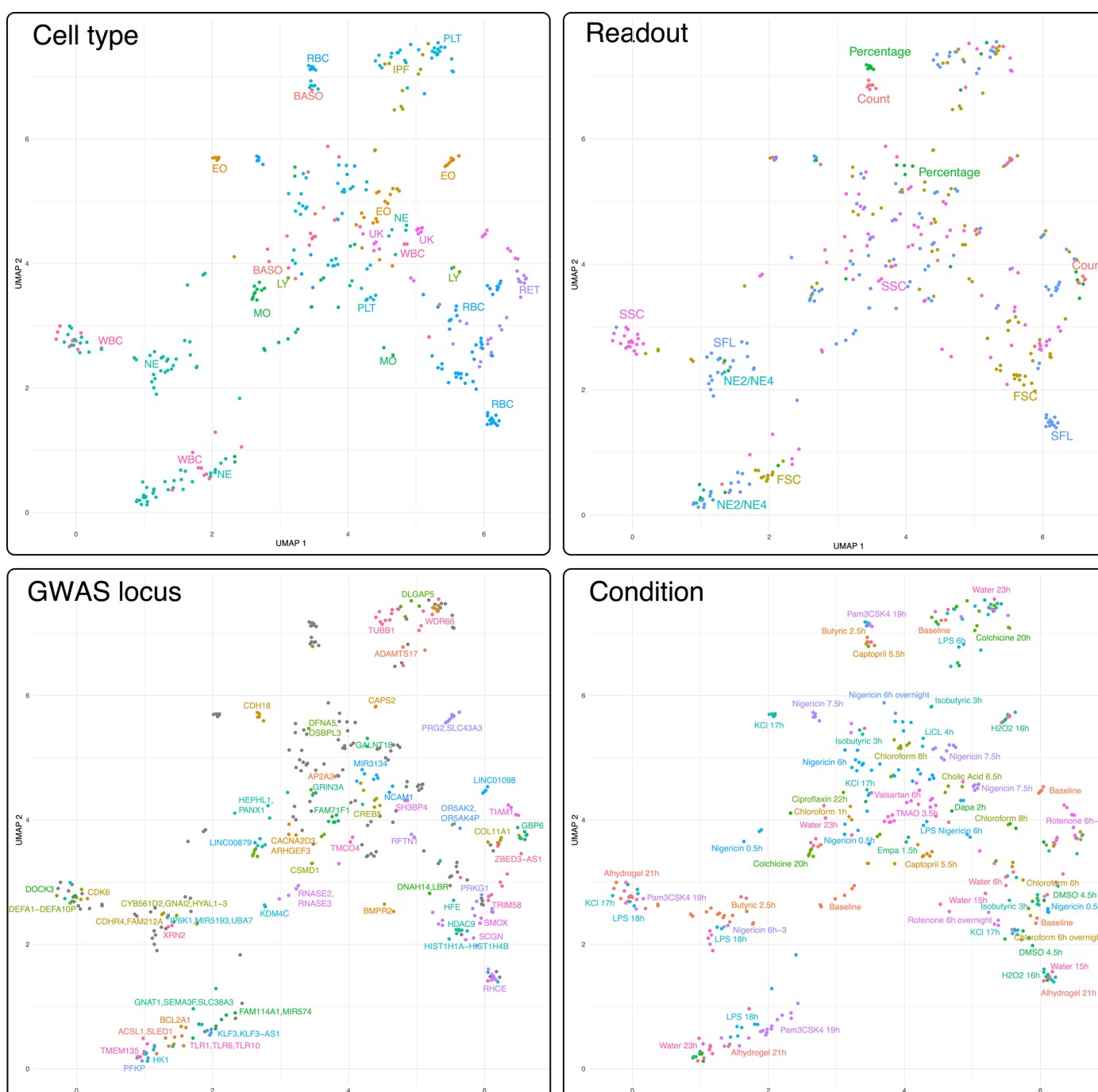

**Extended Data Fig. 3 | Blood readouts with significant genetic associations form clusters based on cell type, readout, perturbation condition and associated genetic loci.** Distance correlations between all pairs of blood trait readouts with significant genetic associations were projected into a 2-dimensional embedding using UMAP. Each trait is assigned a color by the cell type, the type of readout, associated candidate genes at the GWAS locus and the perturbation condition.

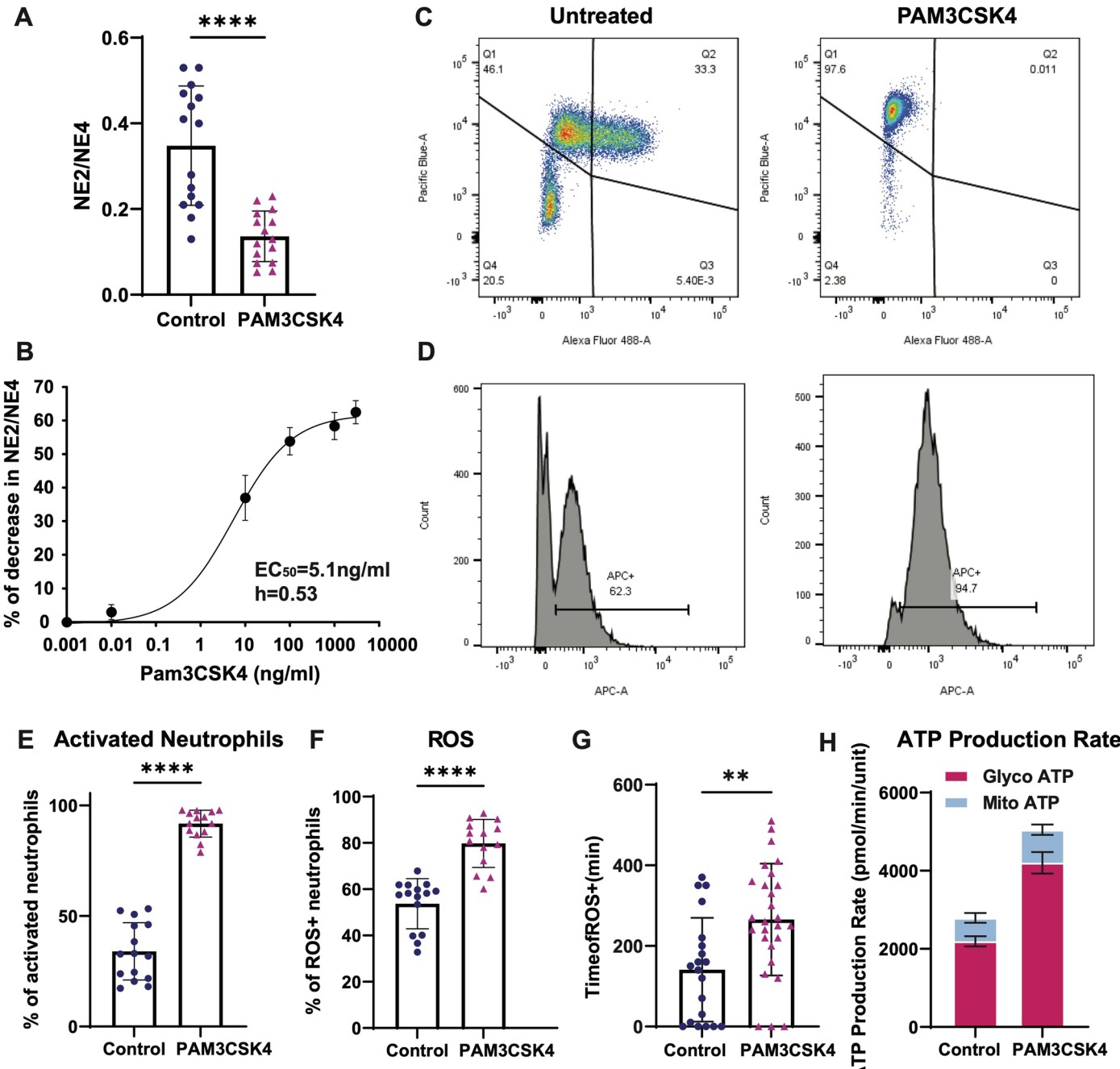

**Extended Data Fig. 4 | Neutrophil response to TLR1/TLR2 ligand Pam3CSK4.**
**a**, NE2/NE4 ratio from Sysmex readout at 17 h in blood incubated with Pam3CSK4, compared to control. $n = 15$ donors' blood samples were examined. Error bars indicate s.d. **b**, Dose-dependent effect of Pam3CSK4 on NE2/NE4 ratio at 17 h post incubation. $n = 9$ donors' blood samples were tested. **c**, Flow cytometry analysis of isolated neutrophils stained with Alexa 488 conjugated CD62L and Pacific blue conjugated CD11b under untreated and Pam3 conditions. **d**, Histogram of CellRox in neutrophils isolated from blood with or without Pam3CSK4 treatment. **e**, Percentage of neutrophil activation under control ($n = 15$ donors' blood samples) or Pam3CSK4 conditions ($n = 14$ donors' blood samples). Error bars indicate s.d. **f**, Percentage of ROS+ neutrophils under control ($n = 15$ donors'

blood samples) and Pam3CSK4 ($n = 14$ donors' blood samples) conditions. Error bars indicate s.d. **g**, Time of neutrophils stay ROS positive under control ($n = 20$ neutrophils from 3 donors) and Pam3CSK4 ($n = 27$ neutrophils from 3 donors) conditions. Error bars indicate s.d. **h**, Seahorse analysis of isolated neutrophils showing ATP produced from the mitochondria and the anaerobic glycolysis pathways in conditions of control and PAM3CSK4 treated. Five replicates were performed for the same donor. Error bars shown indicate s.e.m. of measurements from four donors' blood samples. Paired two-sided $t$-test was used to determine statistical significance in **a**, **e**, and **f**. Unpaired two-sided $t$-test was used in **g**. **$^{**}P < 0.01$, and **$^{****}P < 0.0001$.

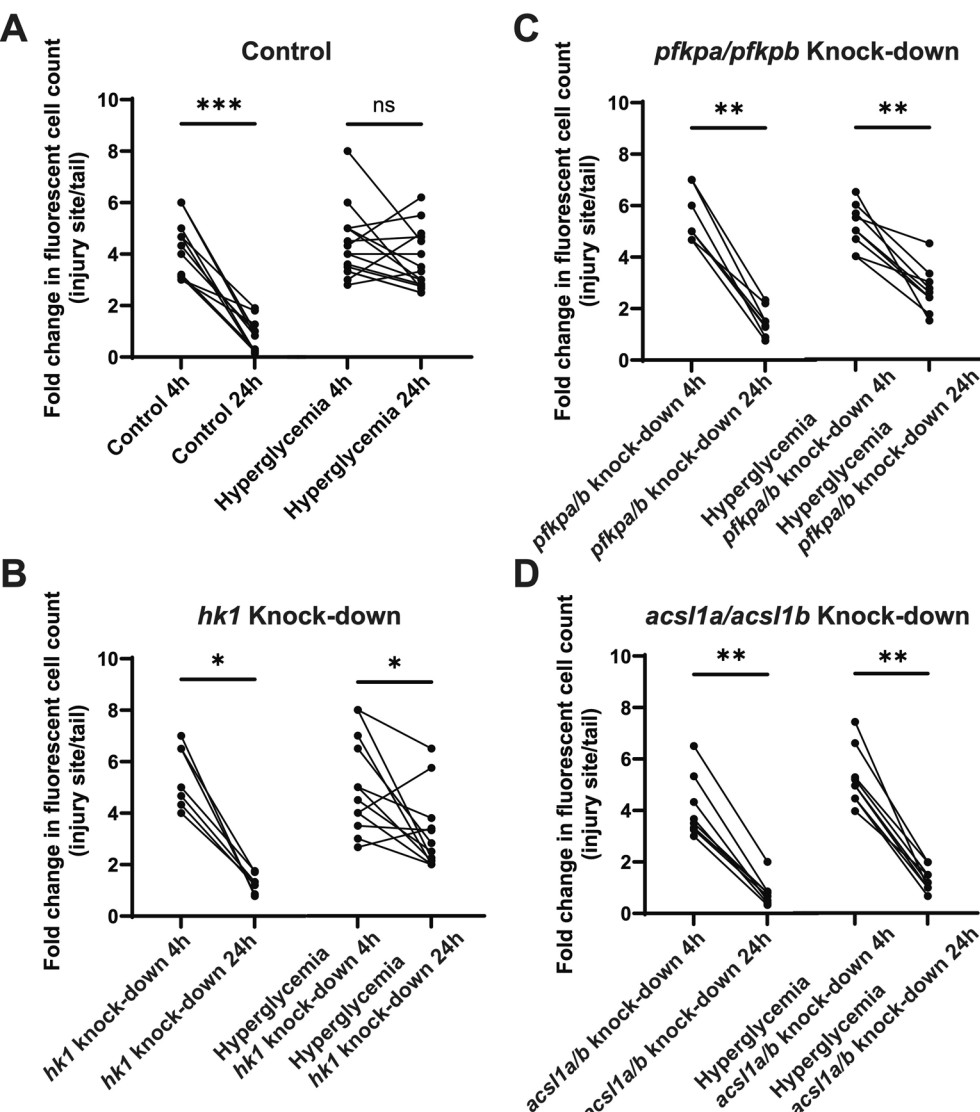

**Extended Data Fig. 5 | CRISPR-Cas9 knockdown of *hk1*, *pfkpa*/*pfkpb*, and *acsl1a*/*acsl1b* in zebrafish promotes neutrophil clearance after tail injury in hyperglycemia. a-d**, Quantification of GFP+ cells at the tail transection site at 4 h and 24 h under control, and hyperglycemia conditions for control Tracer RNA injected (**a**, control *n* = 12, hyperglycemia *n* = 15), *hk1* (**b**, control *n* = 7, hyperglycemia *n* = 12), *pfkpa*/*pfkpb* (**c**, control *n* = 8, hyperglycemia *n* = 9), and *acsl1a*/*acsl1b* (**d**, control *n* = 9, hyperglycemia *n* = 8) gRNA injected zebrafish. Each data point indicates an individual zebrafish. Paired two-sided non-parametric test (Wilcoxon test) was used to determine *P* values. *\**P* < 0.05, *\*\**P* < 0.01, and *\*\*\**P* < 0.001.

## RHCE
### ret Rotenone 6h overnight RET1 CV SFL

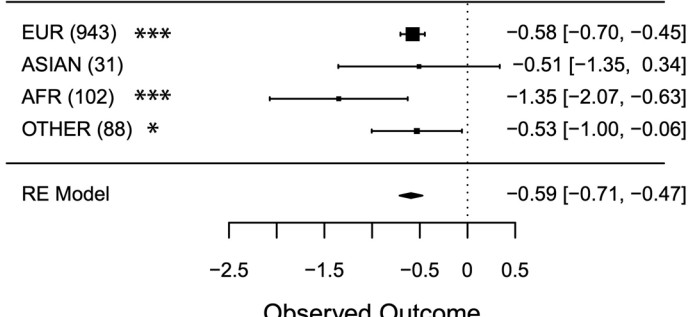

## BCL2A1
### wdf LPS 18h NE4 SD SFL

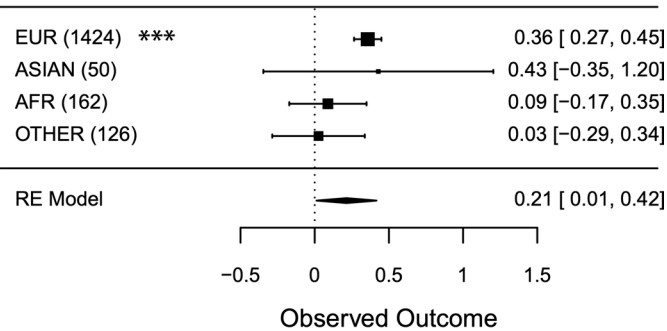

## HK1
### wdf Alhydrogel 21h NE4 SD SFL

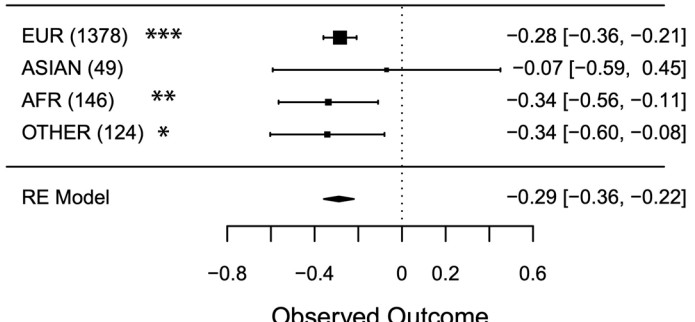

## TLR1
### wdf Pam3CSK4 19h NE1 Med FSC

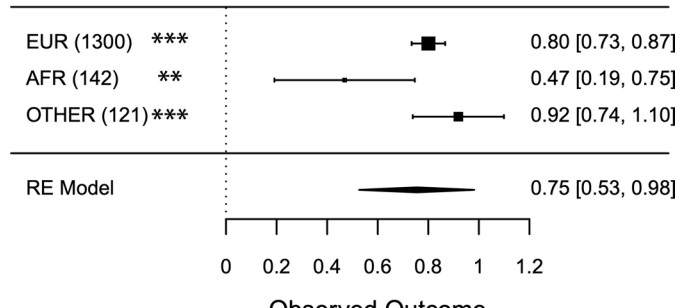

**Extended Data Fig. 6 | Genetic associations for selected traits across different ancestry groups.** Forest plots of ancestry-specific associations for selected lead SNPs. Each plot shows estimated log HR and 95% CI across distinct ancestry groups after harmonizing the effect allele. Genotype counts are shown in parentheses. Asterisks next to each ancestry group denote levels of significance reached: *$P < 0.05$, **$P < 0.01$, ***$P < 0.001$. The size of the squares represents the weight of each study in the meta-analysis. The diamond at the bottom of each forest plot represents the combined effect size and its confidence interval from the multi-ancestry meta-analysis. The SNPs shown are rs644592 (*RHCE*), rs67760360 (*BCL2A1*), rs6480404 (*HK1*) and rs5743618 (*TLR1*).

# Reporting Summary

## Statistics

For all statistical analyses, confirm that the following items are present in the figure legend, table legend, main text, or Methods section.

| n/a | Confirmed | |
|---|---|---|
| ☐ | ☒ | The exact sample size (*n*) for each experimental group/condition, given as a discrete number and unit of measurement |
| ☐ | ☒ | A statement on whether measurements were taken from distinct samples or whether the same sample was measured repeatedly |
| ☐ | ☒ | The statistical test(s) used AND whether they are one- or two-sided *Only common tests should be described solely by name; describe more complex techniques in the Methods section.* |
| ☐ | ☒ | A description of all covariates tested |
| ☐ | ☒ | A description of any assumptions or corrections, such as tests of normality and adjustment for multiple comparisons |
| ☐ | ☒ | A full description of the statistical parameters including central tendency (e.g. means) or other basic estimates (e.g. regression coefficient) AND variation (e.g. standard deviation) or associated estimates of uncertainty (e.g. confidence intervals) |
| ☐ | ☒ | For null hypothesis testing, the test statistic (e.g. *F*, *t*, *r*) with confidence intervals, effect sizes, degrees of freedom and *P* value noted *Give P values as exact values whenever suitable.* |
| ☒ | ☐ | For Bayesian analysis, information on the choice of priors and Markov chain Monte Carlo settings |
| ☒ | ☐ | For hierarchical and complex designs, identification of the appropriate level for tests and full reporting of outcomes |
| ☒ | ☐ | Estimates of effect sizes (e.g. Cohen's *d*, Pearson's *r*), indicating how they were calculated |

*Our web collection on statistics for biologists contains articles on many of the points above.*

## Software and code

Policy information about availability of computer code

| Data collection | Study data of the screening cohort were collected using REDCap v13.1 |
|---|---|
| Data analysis | Plink 1.9 and 2.0, Michigan Imputation Server (docker image) 1.5.7, R 4.2, data.table 1.14.6, ggplot2 3.4.1, survival 3.5-3, survminer 0.4.9, circos 0.69-9, fujiplot, qvalue 2.4.2, PCAtools 2.12.0, fastICA 1.2-3, python 3.9, pandas 1.4.1, numpy 1.19.5, statsmodels 0.13.2, FlowJo 10.8.1, GraphPad Prism 9. Main analysis scripts are available at https://github.com/mxhm/blood_perturbation_gwas. |

For manuscripts utilizing custom algorithms or software that are central to the research but not yet described in published literature, software must be made available to editors and reviewers. We strongly encourage code deposition in a community repository (e.g. GitHub). See the Nature Portfolio guidelines for submitting code & software for further information.

## Data

Policy information about availability of data

All manuscripts must include a data availability statement. This statement should provide the following information, where applicable:
- Accession codes, unique identifiers, or web links for publicly available datasets
- A description of any restrictions on data availability
- For clinical datasets or third party data, please ensure that the statement adheres to our policy

Individual-level data sharing is subject to restrictions imposed by patient consent and local ethics review boards. GWAS summary statistics have been submitted to the GWAS catalog database (study ids: GCST90257015-GCST90257105). PGS are available at https://doi.org/10.6084/m9.figshare.24354235.

# Human research participants

Policy information about studies involving human research participants and Sex and Gender in Research.

| | |
|---|---|
| Reporting on sex and gender | Self-reported sex was used as covariate in clinical association analyses and genetic association analyses of blood readouts, as well as polygenic score analyses. |
| Population characteristics | Population characteristics are described in Extended Data Table 2. |
| Recruitment | Subjects were recruited in accordance with IRB 2019P003155 from multiple phlebotomy clinics in the MassGeneralBrigham hospital system. Subjects were recruited at the time of check-in/registration for their clinical blood draw. Once patients consented, the patient underwent their clinical blood draw first, then the phlebotomist drew blood tubes for the research study. Patients were able to request study staff to fully discuss risks, benefits, and obtain consent at the time of the visit. The study inclusion criteria were age>18, already scheduled to have blood drawn as part of routine clinical care, and ability to provide informed consent. There is no selection bias in recruitment. |
| Ethics oversight | MassGeneralBrigham hospital |

Note that full information on the approval of the study protocol must also be provided in the manuscript.

# Field-specific reporting

Please select the one below that is the best fit for your research. If you are not sure, read the appropriate sections before making your selection.

☒ Life sciences　　　☐ Behavioural & social sciences　　　☐ Ecological, evolutionary & environmental sciences

For a reference copy of the document with all sections, see nature.com/documents/nr-reporting-summary-flat.pdf

# Life sciences study design

All studies must disclose on these points even when the disclosure is negative.

| | |
|---|---|
| Sample size | Samples were collected in multiple phlebotomy clinics over the course of several years. The sample size was not calculated prior to sample collection, as we tried to maximize the number of subjects recruited to maximize power for testing genome-wide associations. We gathered blood samples for 4723 subjects and genotyped 2685 of them (Extended Data Table 2). The number of samples varied for the perturbation conditions from over 641 to 3223 blood samples (Extended Data Table 1). <br> For functional validation studies, sample size calculation was not performed. Sample sizes were chosen based on comparisons to prior similar studies. Notably, for experiments using human blood samples, samples from at least 8 donors were used for each condition/experiment. Although there may be substantial variability among individual donors, the compound treatments used in our study elicited pronounced effects, which were consistently observed with our chosen sample sizes. |
| Data exclusions | Samples were excluded based on pre-established criteria for phenotypic and genetic quality control (online methods). For clinical associations, subjects with organ transplants were excluded due to the effects of immunosuppressant medications. |
| Replication | We analyzed all available samples in the genome wide association studies and no additional datasets are available for replication of the perturbation responses. To reduce risk of false-positive results, we performed quality control on phenotypic and genetic data (online methods). <br> For other experiments involving donor blood samples, each experiment was performed on at least two independent days on independent samples. For experiments involving zebrafish, at least two clutches of zebrafish embryos were used for each experiment. All attempts at replication were successful. |
| Randomization | Randomization is not applicable, since the main study design is a collection of genome-wide association studies of blood cell readouts. The goal of our study is to identify naturally occurring genetic variants associated with measured blood traits within a population, which does not involve assigning participants to different experimental conditions. |
| Blinding | GWAS analyses were not blinded since linking genotype and phenotype information was necessary for statistical analyses. <br> Studies using zebrafish were blinded. Functional studies using human blood samples and compound treatments were not blinded, however, the data were analyzed with the same flow cytometry gating strategy and the analyses were standardized with FlowJo. |

# Reporting for specific materials, systems and methods

We require information from authors about some types of materials, experimental systems and methods used in many studies. Here, indicate whether each material, system or method listed is relevant to your study. If you are not sure if a list item applies to your research, read the appropriate section before selecting a response.

## Materials & experimental systems

| n/a | Involved in the study |
|---|---|
| ☐ | ☒ Antibodies |
| ☒ | ☐ Eukaryotic cell lines |
| ☒ | ☐ Palaeontology and archaeology |
| ☐ | ☒ Animals and other organisms |
| ☒ | ☐ Clinical data |
| ☒ | ☐ Dual use research of concern |

## Methods

| n/a | Involved in the study |
|---|---|
| ☒ | ☐ ChIP-seq |
| ☐ | ☒ Flow cytometry |
| ☒ | ☐ MRI-based neuroimaging |

# Antibodies

| | |
|---|---|
| Antibodies used | Pacific Blue anti-human CD11b antibody (Biolegend, Clone ICRF44, catalog number 3013215) and Alexa Fluor 488 anti-human CD62L antibody (Biolegend, Clone DREG-56, catalog number 304816). |
| Validation | All antibodies used in this study have been validated by the manufacture. They have verified reactivity to human cell/tissue. According to the manufacture, "Each lot of this antibody is quality control tested by immunofluorescent staining with flow cytometric analysis." |

# Animals and other research organisms

Policy information about studies involving animals; ARRIVE guidelines recommended for reporting animal research, and Sex and Gender in Research

| | |
|---|---|
| Laboratory animals | zebrafish (danio rerio) larvae (2-4 day post fertilization) |
| Wild animals | No wild animals were used in the study |
| Reporting on sex | Zebrafish do not have heteromorphic sex chromosomes and their sex are not determined until older than 20-25 day post fertilization. As we used zebrafish larvae in this study, there is no sexual differentiation at this stage. |
| Field-collected samples | No field collected samples were used in the study |
| Ethics oversight | Brigham and Women's Hospital Standing Committee on Animals |

Note that full information on the approval of the study protocol must also be provided in the manuscript.

# Flow Cytometry

## Plots

Confirm that:

☒ The axis labels state the marker and fluorochrome used (e.g. CD4-FITC).

☒ The axis scales are clearly visible. Include numbers along axes only for bottom left plot of group (a 'group' is an analysis of identical markers).

☒ All plots are contour plots with outliers or pseudocolor plots.

☒ A numerical value for number of cells or percentage (with statistics) is provided.

## Methodology

| | |
|---|---|
| Sample preparation | For Sysmex-based measurements, whole blood was collected in 8.5mL ACD tubes (BD 364606). Barcoded sample tubes with patient and perturbation identifiers were aligned and prepared batch-wise, by aliquoting 700ul of whole blood into a grid of 5mL round bottom tubes. All perturbation compounds were added to blood at specified time points and transferred to incubator shakers (39C, 200 RPM). After incubation, tubes were placed in automated sampling racks and profiled using the Sysmex XN-1000.<br>For isolated neutrophil measurements. We used EasySep Direct Human Neutrophil Isolation Kit (#19666, STEMCELL) to isolate neutrophils according to manufacture protocols. Post isolation, neutrophils were resuspended in Tyrode's solution as described previously. To characterize the NE2-like cell population using flow cytometry, neutrophils were isolated from whole blood samples that were incubated at 37°C for 17h, and then labeled with apoptosis indicators, Sytox green (S7020, ThermoFisher Scientific), and R-PE conjugated annexin V (ThermoFisher Scientific). The labeled neutrophils were then subjected to permeabilization using Sysmex WDF Lysercell (Sysmex) and staining with Fluorocell WDF dye (Sysmex). The samples were analyzed 5 minutes post the addition of Flurocell WDF dye.<br>To characterize neutrophil activation and ROS, isolated neutrophils were labeled with Pacific Blue anti-human CD11b antibody (Biolegend, Clone ICRF44) and Alexa Fluor 488 anti-human CD62L antibody (Biolegend, Clone DREG-56). Cells were then subsequentially labeled with CellROX Deep Red Reagent (ThermoFisher Scientific, C10422) at 37°C for 30 minutes. Cells |

were washed and resuspended in staining buffer prior to flow cytometry analyses.

Instrument                     BD FACSymphony, Sysmex XN-1000

Software                       FlowJo v10.8.1

Cell population abundance      Sorting was not employed in this study

Gating strategy                Gates were empirically defined based on densities of measured cells under baseline and treated conditions across all
                               subjects. Gates are shown in Figure 1B.

☒ Tick this box to confirm that a figure exemplifying the gating strategy is provided in the Supplementary Information.

