## [Peer Review File · Nature Genetics]

Peer Review Information

Manuscript Title: Perturbational phenotyping of human blood cells reveals genetically determined latent traits associated with subsets of common diseases

Corresponding author name(s): Calum A. MacRae, Rahul C. Deo, Max Homilius, Wandu Zhu

Reviewer Comments & Decisions:

Decision Letter, initial version:

9th May 2023

Dear Dr. Homilius,

Your Article "Perturbational phenotyping of human blood cells reveals genetically determined latent traits associated with subsets of common diseases" has been seen by three referees. You will see from their comments below that, while they find your work of potential interest, they have raised substantial concerns that must be addressed. In light of these comments, we cannot accept the manuscript for publication at this time, but we would be interested in considering a suitably revised version that addresses the referees' concerns.

We hope you will find the referees' comments useful as you decide how to proceed. If you wish to submit a substantially revised manuscript, please bear in mind that we will be reluctant to approach the referees again in the absence of major revisions.

To guide the scope of the revisions, the editors discuss the referee reports in detail within the team, including with the chief editor, with a view to identifying key priorities that should be addressed in revision, and sometimes overruling referee requests that are deemed beyond the scope of the current study. In this case, we ask that you address all technical queries related to the study design and analysis strategy with clarifications and additional analyses where appropriate and revise the presentation for clarity throughout. We hope you will find this prioritized set of referee points to be useful when revising your study. Please do not hesitate to get in touch if you would like to discuss these issues further.

If you choose to revise your manuscript taking into account all reviewer and editor comments, please highlight all changes in the manuscript text file. At this stage we will need you to upload a copy of the manuscript in MS Word .docx or similar editable format.

We are committed to providing a fair and constructive peer-review process. Do not hesitate to contact

us if there are specific requests from the reviewers that you believe are technically impossible or unlikely to yield a meaningful outcome.

*2) If you have not done so already, please begin to revise your manuscript so that it conforms to our Article format instructions, available [here](http://www.nature.com/ng/authors/article_types/index.html). Refer also to any guidelines provided in this letter.

[redacted]

If you wish to submit a suitably revised manuscript, we would hope to receive it within 3-6 months. If you cannot send it within this time, please let us know. We will be happy to consider your revision so long as nothing similar has been accepted for publication at Nature Genetics or published elsewhere. Should your manuscript be substantially delayed without notifying us in advance and your article is eventually published, the received date would be that of the revised, not the original, version.

Nature Genetics is committed to improving transparency in authorship. As part of our efforts in this direction, we are now requesting that all authors identified as 'corresponding author' on published papers create and link their Open Researcher and Contributor Identifier (ORCID) with their account on the Manuscript Tracking System (MTS), prior to acceptance. ORCID helps the scientific community achieve unambiguous attribution of all scholarly contributions. You can create and link your ORCID from the home page of the MTS by clicking on 'Modify my Springer Nature account'. For more information, please visit www.springernature.com/orcid.

Thank you for the opportunity to review your work.

Sincerely,
Kyle

Kyle Vogan, PhD
Senior Editor
Nature Genetics
<https://orcid.org/0000-0001-9565-9665>

Referee expertise:

Referee #1: Genetics, blood cell traits, cardiovascular diseases

Referee #2: Genetics, blood cell traits, cardiovascular diseases

Referee #3: Genetics, blood cell traits, cardiovascular diseases

Reviewers' Comments:

Reviewer #1:
Remarks to the Author:

In this manuscript, Homilius and colleagues test the interesting hypothesis that many endophenotypes relevant to disease etiology cannot be measured (and therefore studied) at baseline (without stimulations). This is not novel, and several genetic studies have previously looked at a limited number of phenotypes upon stimulations, especially in the context of eQTL studies. The scale of the experiment by Homilius and colleagues is however very impressive. They deeply characterized blood cells from up to 2600 donors using various cell-based measurements and stimuli. Then, they used these measurements to test for correlations with disease phenotypes and to perform GWAS. Their results are intriguing: (1) many such endophenotypes are associated with diseases (or clinically relevant traits), (2) many genetic variants associate with these latent phenotypes, but not with the simpler quantitative values obtained from routine CBC, and (3) polygenic scores based on these latent phenotypes predict diseases.

One of the main challenges of this study is to explain how these endophenotypes may relate to human diseases, especially because these variables have never been described before (i.e. it is unclear what biology they capture). To partly address this, the authors focused on a new population of neutrophils, and showed how these cells may modulate inflammatory responses in hyperglycemic conditions. Despite this success story, it is unclear how useful most of these phenotypes will be to improve our understanding of disease pathophysiology.

I have the following questions/comments:

1. The description of the results could be significantly improved. The large heatmaps are not particularly useful, nor is Figure 2 which is hard to understand. The information found in the Supplementary Tables needs to be better described as well. For instance, all headers should be defined. Currently, it is hard to know what these values are, which will preclude the use of the data by others.
 2. Have you done replicates on blood samples from the same participants to assess phenotype reproducibility? If you have done replicates, were the samples obtained from the same blood draw, or from blood samples collected at different times?
 3. In Table 1 and Figure 2, how were the candidate genes selected? Based on physical distance? The results don't implicate genes per se, so I suggest that the authors explain their selection criteria and emphasize that these are candidate causal genes.
 4. Related to the point above, what are the evidences that the SNPs at PFKP, HK1 and ACSL1 modulate the activity of these genes? There is a sentence on page 7 ("we found SNPs leading to upregulation of HK1 and PFKP expression"), but I don't know what supports this claim.
 5. At the top of page 4, the authors make the statement that their experimental design allows them to identify associations which would require much larger sample size when focusing on CBC parameters at baseline. I think that this is probably true, but it would be nice if they could quantify. One idea would be to take the variants from Table 1 that have previously been associated with CBC traits and compare effect sizes (or variance explained) (between CBC traits and the latent phenotypes described here).
 6. There is no replication. And association results are not corrected for the number of tested phenotypes. Taken together, there are likely false positive associations among their results. The authors should probably discuss this limitation in their Discussion.
 7. Is the data presented in Figure 3 robust to CBC adjustments? For instance, on page 4, it says that QTc is correlated with ret LPS 18h RBC1 SD FSC. Does this correlation remain if you correct for MCV? Same question for the ratio NE2/NE4 with cardiometabolic diseases if you correct for NEU count?
 8. Figure 5. How do the violin plots in the bottom row relate to the flow cytometry profiles in the top row? I think that the top row is somewhat a subtraction of flow cytometry signals, but I do not understand which of the ovals represent the NE2 or NE4 populations.
 9. Are the blood polygenic scores associated with diseases (Fig. 7) correlated with other polygenic scores specifically tailored for the same diseases? For instance, is the polygenic score for ret KCl 17h RET2 SD SSC associated with heart failure also correlated with a polygenic score specific to heart failure? Is a polygenic score for NE2/NE4 better than a CKD-specific score at stratifying diabetic patients with CKD? I am trying to determine if these latent variable blood phenotypes capture something more than the "standard" polygenic scores (which would be really exciting). You could try scores for some of the most promising phenotypes using data from the polygenic score catalog.
- Minor
10. Figure 4H is not described in the main text.

Reviewer #2:
Remarks to the Author:

Homilius and colleagues applied Sysmex flow cytometry with varying dyes to measure PBMC cell characteristics/populations (in up to $n \sim 4,700$) before and after application of stimuli (perturbations) and searched for genetic alleles (in up to $n \sim 2,600$ individuals) associated with these intermediate phenotypes. Polygenic risk scores (PRS) were derived for 327 readout phenotypes and these were checked for association with 20 clinical outcomes. Particular focus is given to an inflammation stimulus provoked population with characteristics of neutrophils that pro-inflammatory and anti-apoptotic linked to GWAS-significant variants in loci including TLR1, PFKP and HK1. These findings are followed up with traditional flow cytometry markers to characterize the populations and directed experiments including in zebrafish. While not the first effort to apply perturbation approaches to genetic mapping of cellular phenotypes (well trod in model organisms where manipulation of diet, sleep, social interactions and treatments have been applied in relation to OMICs/genetics, but applied more sparingly in human studies), the scale of the current effort is a major advance in this area. I have a number of comments and questions on the work to be clarified.

Major comments

1. The authors compare their cellular results to CBC GWAS from Vuckovic et al., and from an older paper from Astle et al. While they cite Chen et al., which appeared in the same issue of Cell as Vuckovic et al., they do not compare results with Chen et al. In order to claim novelty of your loci, you should compare to the largest available study (which also matched your trans-ethnic design unlike Vuckovic and Astle which were European ancestry only). Chen et al. included $\sim 750,000$ samples.
2. I struggle the most with the application of the PGS to outcomes. Kaplan-Meier / Cox regression should be applied from a point at which someone enters a study at baseline. You show "follow-up" time that in some cases span 50+ years of event surveillance. I believe what you did was retrospectively take the first instance of any event as a baseline age and then assume that all others had been followed "without events" for this many years if you did not find an instance of disease in their medical records. However, this approach would seem quite flawed as presumably you do not have 50 years' worth of complete medical records on every individual in the study. I realize you censored people with known past events, but are your records reliable enough to cover these full age spectra in all individuals? Second, please clarify your use of FDR. Was this computed only on a per clinical outcome, or globally across clinical outcomes (there should have been 20×327 readouts = 6,540 tests)? I suspect it was the former, which seems a liberal threshold. Third, the validation results in the Screening cohort (for outcomes with ≥ 20 cases) should be presented an Extended data table. Instead of a nominal p-value threshold you should apply a significance threshold either at an FDR rate or a Bonferroni threshold ($0.05/31$) to declare significance in validation. Fourth, there are 26 ICD-10 code listings listed in Extended Data Table 3, rather than 20. Were some of these grouped together (if so, that should be made clear), or did some not reach 20 cases? Perhaps the number of cases should be listed in the Table?
3. In general, those conditions that tested the largest sample size yielded the most significant results (Extended Data Table 1), whereas some of the drug stimuli, TMAO, etc. did not. It may reflect the variability in power among the tested conditions given the wide range in sample size and this may be

worth comment in terms of potential False Negatives here, and the design of future studies toward larger samples. I think you should make some comment on the limitations of your varied power across the experiments since you have nearly a 10-fold range in sample sizes with different components.

4. One of the problems that arises in flow cytometry is the migration of analytic gates across time which can be due to a variety of factors including changes in antibody batches/clones/manufacture, fluctuation in instruments requiring calibration and voltage settings, changes in detectors, fluctuating or inconsistent pressure within the system, etc. You mention that you initially tested the reproducibility of the various assays and conditions, but you do not present any of this data. Since this assay approach is relatively new knowledge in large PBMC samples, it would be helpful if you presented results on the reproducibility and other QC metrics in a Supplement. Do you have any repeated blood donors over the course of the study that might address the stability of the analytic gates within samples? You present a merged view of the data in a heatmap in Extended Data Figure 1. This is somewhat informative. There is also a movement in Flow cytometry to present multiple representative Gating plots from independent samples (akin to the prior movement to show several Western Blot images or other gel images). It might be helpful to show several independent samples gating for the various tests and conditions in a Supplement.

5. I am curious about several loci in Figure 2 that seem to show “pleiotropic” patterns across stimuli but are not commented on by the authors. These often seem to show associations with Baseline, Water, KCl, LPS, and/or other conditions. Why do you not focus at all on these – are they suspected to be generalized cellular QTLs that are not very condition-specific, or could they be loci that are sensitive and important for responses to many stimuli? What does a Water QTL represent – some sort of osmotic or pH change sensitive response?

MINOR COMMENTS

1. The authors apply a standard/older GWAS threshold of $5E-08$ despite testing many traits. They also utilize an older imputation panel (1000G) than current panels such as HRC and TOPMed that provide better coverage and imputation. Personally, this is fine with me as a screening approach for further downstream uses (as in this case) [I think it would have been inadequate had the study just stopped there]. Other reviewers and readers may feel differently as the GWAS field has generally moved toward more conservative thresholds as # of variants tested and # of traits per study have both increased. Again, from my perspective this is not vital to address, but I think I would be remiss not to raise it as a potential concern.

2. Given recent publication on use of race and ancestry in genetics (e.g., NAS and TOPMed publication), please be more specific about how ancestry was defined in your sample.

3. Is 39C a typo, or perturbations done above physiological temp for a reason? Please explain.

4. Was it a single Sysmex instrument or multiple ones? If multiple, did you analyze possible instrument variability?

5. Extended Table 3 – K85.0 code is listed twice, and unclear if it really differed from K85

6. What measure of LD >0.50 for clumping (R^2 , D , D')?

7. Please define what padj means at first use (i.e., FDR threshold or whatever it is if it varies across analyses)
8. In Extended Data Fig. 7 for RBC trait, no data was plotted in my version for Age vs. RBC1 Med SFL. I'm not sure if this is an author omission or a problem with the journal processing or PDF download on my end.
9. In the Introduction you "utilized live circulating" cells. This is not exactly accurate. You removed the cells from circulation before assaying as is typically the case for most such assays (unless in vivo imaging is used, or a proxy circulatory system used). So I suggest modifying the phrase slightly here.
10. In the Introductory description, which is overall very well written, in circulation you fail to mention cell-free DNA, microparticles and extra-cellular vesicles which are other mechanisms for biological readouts or cell-cell communication.
11. In addition to the HFGP project, there is a corollary study type which is probing platelet functional responses to agonist stimuli (the largest/most recently WGS in n~3,800 Keramati et al. Nat Commun PMID 34131117 and Rodriguez et al. AJHG PMID 32649856 which found a large QTL associated with venous and arterial thrombosis).
12. Ref. 19 has been on BioRxiv server for 3+ years. I'm not sure if there is an age where a preprint becomes less appropriate to cite but this seems old to me. Perhaps this is a journal question, or perhaps the article will be published and citation changed by the time this article might go to press.

Reviewer #3:
Remarks to the Author:

The authors utilize an innovative GWAS-based framework to study various endophenotypes related to a broad range of chemical and pharmacologic perturbations of blood cells measured using a routine hematology analyzer. The combinations of cell types, readouts, fluorophore, and ~40 chemical stimuli generated several thousand parameters per subject in up to 3300 subjects. Using this approach, 100 loci were associated with at least two blood cell-perturbation readouts, including 23 perturbation loci not previously reported. The authors then performed focused investigation and further characterization of a particular subpopulation (NE2) of apoptotic neutrophils that appear following inflammatory stimuli whose presence is inversely related to extent of neutrophil activation and pro-inflammatory response and also inversely associated with risk of clinical cardiometabolic phenotypes in the same MGB data set. These included experiments on neutrophil metabolism and zebrafish related to candidate genes HK1 and PFKP. The authors also generated polygenic scores for the evoked blood cell endophenotypes and assessed their relationship to clinical outcomes in MGB.

Comments:

1. The subjects were recruited from phlebotomy clinics of a tertiary hospital system, and therefore may be enriched for specific conditions or diagnoses, rather than being representative of the general population. Therefore, it would be useful to describe or summarize the clinical, demographic, and lifestyle characteristics of the study participants. Were individuals excluded for conditions known to

strongly influence blood cell parameters (e.g., primary hematologic or bone marrow disorders, cancer, myelosuppressive chemotherapy) or acute inflammatory conditions? This would be important to assess how generalizable the findings might be to the general population.

2. Fig 1D – what is the X-axis?

3. It is stated that non-European individuals were excluded from GWAS. What is the race/ethnicity breakdown of the participants? How many non-European individuals were excluded? Is the final # included in the GWAS = 2685? Given that there are known differences in the genetic architecture of blood cell counts and inflammatory responses between different race/ethnicities, how generalizable are the findings expected to be to non-Europeans?

4. The conventional genome-wide significance threshold of 5×10^{-8} seems liberal, given that thousands of different phenotypes (albeit some of them correlated) were tested. The effective number of uncorrelated traits could be estimated and used to correct for multiple testing.

5. I am quite confused about the description of the Results related to Extended Data Fig 5. The paragraph discusses an increase in NE2/NE4 ratio with Pam3CSK4 treatment and the relationship of higher NE2/NE4 to neutrophil proinflammatory responses. Which appears to be the exact opposite of what the data in the Figure shows.

6. The authors state “The lead SNPs we identified cause upregulation in HK1 and PFKP expression and decreased NE2/NE4 ratio, suggesting reduced neutrophil apoptosis.” What are the data that suggest that these SNPs cause upregulation of HK1 and PFKP?

7. The authors report that polygenic scores for evoked blood cell readouts predict occurrence of various clinical outcomes within the same MGB biobank data set. Since only genotype data are required to assess the association of PGS with clinical outcomes, it would be helpful to see some of these PGS – clinical trait associations replicated in independent data sets with genome-wide genotyping available.

Author Rebuttal to Initial comments

Responses to Reviewers' Comments

Reviewer #1:

Remarks to the Author:

In this manuscript, Homilius and colleagues test the interesting hypothesis that many endophenotypes relevant to disease etiology cannot be measured (and therefore studied) at baseline (without stimulations). This is not novel, and several genetic studies have previously looked at a limited number of phenotypes upon stimulations, especially in the context of eQTL studies. The scale of the experiment by Homilius and colleagues is however very impressive.

They deeply characterized blood cells from up to 2600 donors using various cell-based measurements and stimuli. Then, they used these measurements to test for correlations with disease phenotypes and to perform GWAS. Their results are intriguing: (1) many such endophenotypes are associated with diseases (or clinically relevant traits), (2) many genetic variants associate with these latent phenotypes, but not with the simpler quantitative values obtained from routine CBC, and (3) polygenic scores based on these latent phenotypes predict diseases.

One of the main challenges of this study is to explain how these endophenotypes may relate to human diseases, especially because these variables have never been described before (i.e. it is unclear what biology they capture). To partly address this, the authors focused on a new population of neutrophils, and showed how these cells may modulate inflammatory responses in hyperglycemic conditions. Despite this success story, it is unclear how useful most of these phenotypes will be to improve our understanding of disease pathophysiology.

We thank the reviewer for their encouraging comments and constructive suggestions. We agree that a central challenge is to link endophenotypes with disease pathophysiology. We not only associated the evoked traits (endophenotypes) with traditional disease phenotypes within the EHR, but we also used the genetics of these endophenotypes and related PGS to confirm associations between the underlying molecular pathways governing the endophenotypes and clinical disease phenotypes, thus further reinforcing potential causal links with disease pathophysiology. We used the neutrophil study as an initial proof of concept, but we anticipate that many of the associations we have identified will be explored in future investigations. We have addressed each of the comments in detail as shown below.

I have the following questions/comments:

1. The description of the results could be significantly improved. The large heatmaps are not particularly useful, nor is Figure 2 which is hard to understand. The information found in the Supplementary Tables needs to be better described as well. For instance, all headers should be defined. Currently, it is hard to know what these values are, which will preclude the use of the data by others.

Thank you for your suggestions on improving the readability of our paper. To enhance the interpretability of the heatmaps for clinical associations, we have incorporated independent component analyses of the blood endophenotypes and clinical traits

(New Fig. 3C and Fig. 7C). These analyses demonstrate that combinations of blood endophenotypes effectively separate different diseases, and also cluster diseases with linked pathophysiology. Additionally, we have plotted projections of multiple exemplar endophenotypes alongside the associated diseases, highlighting their directionality of change and disease associations (arrows in Fig. 3C and Fig. 7C). We believe the new analyses complement the heatmaps and make the data more visually interpretable. Furthermore, we have included Figure 3A to illustrate population distributions of raw blood endophenotypes between cases and controls for four diseases as examples.

For Figure 2, we kept the Circos plot, as it is a commonly used format to present GWAS data for multi-trait analyses (Kanai et al., Nat. Genet. 2018, Sinnott-Armstrong et al., Nat. Genet. 2021). The large number of traits for each cell type limits the level of detail at which we are able to plot associations between loci and traits. We introduced varying dot sizes in the Circos plot, representing the respective p-values. To complement the Circos plot, we have introduced a new Figure 2B. This figure provides a comparison of beta coefficients observed under perturbed conditions, baseline conditions, and those reported in prior studies based on baseline readouts. Specifically, we focused on associations reported for 6 example genes. The findings from this analysis demonstrated that the effect sizes for associations observed under perturbed conditions can be substantially larger than those observed under baseline conditions. These data highlight the potential of chemical or other perturbations to identify novel loci with substantial effects.

We have also improved the descriptions in the headers for the supplementary tables.

2. Have you done replicates on blood samples from the same participants to assess phenotype reproducibility? If you have done replicates, were the samples obtained from the same blood draw, or from blood samples collected at different times?

We have performed replicates to test reproducibility, both technical replicates from the same blood draw and subject replicates for blood collected at different times from the same subjects (over the course of months). We have now added Extended Data Fig. 2 presenting gate definitions and cell counts in the WDF channel, which has the largest number of different cell types in our dataset. As an example, we visualized replicated measurements from the same blood draw for 4 randomly selected subjects in Extended Data Fig 2A. We also had a small number of subjects who repeatedly donated blood to our study over time. We show WDF channel data in Extended Data Figure 2B for these individuals, together with the distribution of a subset of readouts over time in Extended Data Figure 2C for the same individuals. This figure demonstrates that our measurements and gates are reproducible over time.

3. In Table 1 and Figure 2, how were the candidate genes selected? Based on physical distance? The results don't implicate genes per se, so I suggest that the authors explain their selection criteria and emphasize that these are candidate causal genes.

We added additional descriptions in the Methods section for how candidate genes were selected:

“[...] we clumped all significant variants using plink with LD $r^2 > 0.50$, physical distance <250kb between clumped variants, and at least two independent hits from different traits for each clumped region. We used the variant with the smallest association p-value across all measured traits for a given region as the lead variant. The following command was used for clumping and gene range annotations: *plink --clump-range glist-hg19 --clump-p1 0.00000005 --clump-p2 0.00000005 --clump-r2 0.50 --clump-kb 250 --clump-replicate --clump {trait_files}*. This command also annotated associated regions using gene range lists provided by the authors of plink2 (<https://www.cog-genomics.org/static/bin/plink/glist-hg19>). If multiple genes were present for a given location, we used L2G pipeline from OpenTargets Genetics to identify likely candidates. We prioritized candidate genes in the following order: coding variants, variants in introns, distance to TSS. If there was no clear evidence for a subset of candidates, we reported the full list from the Plink gene annotation step.” Pg. 17 Line 20-32

Importantly, our approach generates directly accessible large effect size traits that can allow experimental testing of potential causal relationships between gene effects and the relevant functional metrics. By design, these features allow rigorous experimentation at scale in human cells or in animal models to efficiently define the causal gene or genes for each phenotype through empiric testing of any number of potential candidates. As an example, we have now included additional data (new Extended Data Fig. 8) using CRISPR-Cas9 to target zebrafish orthologs of *HK1*, *PFKP*, and *ACSL1*. These genetic manipulations accelerate the in vivo resolution of neutrophil accumulation at an injury site under hyperglycemia conditions. These results provide additional experimental validation that these candidate genes are causal for the neutrophil survival phenotype we observed.

4. Related to the point above, what are the evidences that the SNPs at PFKP, HK1 and ACSL1 modulate the activity of these genes? There is a sentence on page 7 (“we found SNPs leading to upregulation of HK1 and PFKP expression”), but I don't know what supports this claim.

We apologize for not including the references for this statement. We stated this based on the eQTLs for the lead SNPs for these two genes. We have now included the references for the eQTL data. “The lead SNPs we identified for HK1 and PFKP were previously found to cause upregulation in these two genes expression

(rs6480404 eQTL for HK1 in neutrophil (Chen et al., Cell, 2016): $\beta=0.178$, $p=4e-16$; rs34538474 eQTL for PFKP in blood: $\beta=0.457$, $p=3.3e-310$) (Võsa & Claringbould et. al., Nature Genetics, 2021).” Pg. 7 Line 5-7.

5. At the top of page 4, the authors make the statement that their experimental design allows them to identify associations which would require much larger sample size when focusing on CBC parameters at baseline. I think that this is probably true, but it would be nice if they could quantify. One idea would be to take the variants from Table 1 that have previously been associated with CBC traits and compare effect sizes (or variance explained) (between CBC traits and the latent phenotypes described here).

Thank you for this excellent suggestion. We have now included a comparison of effect sizes between baseline CBC traits and perturbed latent phenotypes from our cohort (New Fig. 2B). Additionally, we have added a gene-level comparison with effect sizes from previously published CBC traits, which were derived from much larger cohorts (New Fig. 2B). The comparison reveals that the effect sizes of the latent perturbed blood traits can be over a log order larger than those of baseline CBC traits, demonstrating the value of perturbational profiling for the discovery of large-effect variants.

6. There is no replication. And association results are not corrected for the number of tested phenotypes. Taken together, there are likely false positive associations among their results. The authors should probably discuss this limitation in their Discussion.

We have now added a limitation section in our discussion regarding false positives. “... we utilized a conventional threshold of $5E-8$ for genetic association without adjusting for the number of tested phenotypes. While we tested thousands of phenotypes, many of those are correlated, and we estimated that approximately 350 traits were independent. However, this estimation may not be accurate, since it was based on a subset of subjects with the highest number of common perturbational conditions measured. To reduce the false discovery rate, we reported significant associations only when at least two independent traits were linked to the clumped region. Nevertheless, due to our relatively small sample size and approach, some false positive associations may still be present in the results.” Pg. 11 Line 1-8.

7. Is the data presented in Figure 3 robust to CBC adjustments? For instance, on page 4, it says that QTc is correlated with ret LPS 18h RBC1 SD FSC. Does this correlation remain if you correct for MCV? Same question for the ratio NE2/NE4 with cardiometabolic diseases if you correct for NEU count?

We intentionally did not normalize the perturbational readouts to baseline CBC traits because it was difficult to determine the exact baseline trait against which to

normalize each perturbed parameter and choosing the wrong parameter might potentially skew the data. Each of our perturbed conditions was measured at a different time point from our baseline measurement, so the time difference may make some of the traits, for example, neutrophil count, not comparable between baseline and perturbed conditions. We did introduce some ratiometric indices, such as NE2/NE4, which indicates NE2 gated (apoptotic) neutrophils over total neutrophils (NE4), to account for the effects of neutrophil count differences among individuals. Overall, our strategy appears to be effective, considering we do see many additional disease associations, sometimes with opposite effects, in the perturbed conditions when compared to baseline, suggesting that we are not simply detecting baseline CBC traits that are linked to diseases.

8. Figure 5. How do the violin plots in the bottom row relate to the flow cytometry profiles in the top row? I think that the top row is somewhat a subtraction of flow cytometry signals, but I do not understand which of the ovals represent the NE2 or NE4 populations.

We apologize that the figure was not explained clearly. The top row panels are the flow cytometry profiles in the WDF channel comparing subjects with the homozygous major and minor alleles. We have now added labels indicating the NE2 and NE4 gates to Fig. 5A (top row) to highlight the relevant cell populations. The lower violin plots (now Fig. 5B) show the NE2/NE4 ratio comparing different genotypes, which correspond to NE2/NE4 ratios calculated from the respective flow cytometry profile. We have now modified the Fig. 5 figure legends to explain this better.

9. Are the blood polygenic scores associated with diseases (Fig. 7) correlated with other polygenic scores specifically tailored for the same diseases? For instance, is the polygenic score for ret KCl 17h RET2 SD SSC associated with heart failure also correlated with a polygenic score specific to heart failure? Is a polygenic score for NE2/NE4 better than a CKD-specific score at stratifying diabetic patients with CKD? I am trying to determine if these latent variable blood phenotypes capture something more than the “standard” polygenic scores (which would be really exciting). You could try scores for some of the most promising phenotypes using data from the polygenic score catalog.

This is an interesting question that we believe is worth exploration. The limited number of subjects in our screening cohort constrained certain analyses, such as genetic correlation computations, usually requiring larger sample sizes for robust estimates. In our time-to-event analyses in MGB and UK Biobanks, previously published PRS for CKD and T2D generally showed stronger predictive power than our single blood- perturbation response PGSs (data not shown). However, our study intentionally used a simple PRS model to demonstrate that genetic signals for blood endophenotypes can stratify these common diseases. We did not fine-tune PRS

scores for cross-trait prediction, nor explore multi-PRS models that have been shown to improve risk stratification in other studies (e.g., Sinnott-Armstrong 2021, Nature Genetics), but with larger cohorts incorporating perturbational phenotyping we anticipate that this will be readily feasible. We believe that by incorporating these approaches, we can significantly improve our PGS model performance for disease stratification, which we plan to pursue in future investigations.

Minor

10. Figure 4H is not described in the main text.

We apologize for the oversight in not citing the figure in the main text. We have now added the description of Figure 4H: “In addition, we assessed ROS generation using CellROX and quantified the percentage of ROS-positive neutrophils for each subject (Fig. 4H). “ Pg. 6, Line 18-19.

Reviewer #2:

Remarks to the Author:

Homilius and colleagues applied Sysmex flow cytometry with varying dyes to measure PBMC cell characteristics/populations (in up to $n \sim 4,700$) before and after application of

stimuli (perturbations) and searched for genetic alleles (in up to $n \sim 2,600$ individuals) associated with these intermediate phenotypes. Polygenic risk scores (PRS) were derived for 327 readout phenotypes and these were checked for association with 20 clinical outcomes. Particular focus is given to an inflammation stimulus provoked population with characteristics of neutrophils that pro-inflammatory and anti-apoptotic linked to GWAS- significant variants in loci including TLR1, PFKP and HK1. These findings are followed up with traditional flow cytometry markers to characterize the populations and directed experiments including in zebrafish. While not the first effort to apply perturbation approaches to genetic mapping of cellular phenotypes (well trod in model organisms where manipulation of diet, sleep, social interactions and treatments have been applied in relation to OMICs/genetics, but applied more sparingly in human studies), the scale of the current effort is a major advance in this area. I have a number of comments and questions on the work to be clarified.

We thank the reviewer for their interest and for their helpful comments which have greatly improved our manuscript. Please see our detailed responses below.

Major comments

1. The authors compare their cellular results to CBC GWAS from Vuckovic et al., and from an older paper from Astle et al. While they cite Chen et al., which appeared in the same issue of Cell as Vuckovic et al., they do not compare results with Chen et al. In order to claim novelty of your loci, you should compare to the largest available study (which also matched your trans-ethnic design unlike Vuckovic and Astle which were European ancestry only). Chen et al. included ~750,000 samples.

We initially did not include this study for comparison, because it is a multiethnic study. However, we understand the reviewer's concern and compared our findings to Chen et al. Our comparison did not reveal any additional candidate gene overlap, compared to Vuckovic et al/ Astle et al. Thus, the number of novel genetic loci we identified remains unchanged (23 new genetic loci).

Although Chen et al., included the largest sample size, they only reported 71 additional variants compared to Vuckovic et al., revealing a limited number of novel genes. The number of distinct blood cell traits measured in Astle et al. and Vuckovic et al. (36 and 29) was higher than in Chen et al. (15). This observation also emphasized an important point: the limitations of GWAS studies for specific phenotypes no longer hinge solely on sample size once a certain threshold is achieved. Instead, the dynamic range and precision of phenotyping play a more substantial role in shaping gene detection. We have now included a citation of Chen et al., but kept Astle and Vuckovic et al. for the primary comparisons.

2. I struggle the most with the application of the PGS to outcomes. Kaplan-Meier / Cox regression should be applied from a point at which someone enters a study at baseline. You show "follow-up" time that in some cases span 50+ years of event surveillance. I believe what you did was retrospectively take the first instance of any event as a baseline age and then assume that all others had been followed "without events" for this many years if you did not find an instance of disease in their medical records. However, this approach would seem quite flawed as presumably you do not have 50 years' worth of complete medical records on every individual in the study. I realize you censored people with known past events, but are your records reliable enough to cover these full age spectra in all individuals? Second, please clarify your use of FDR. Was this computed only on a per clinical outcome, or globally across clinical outcomes (there should have been 20 X 327 readouts = 6,540 tests)? I suspect it was the former, which seems a liberal threshold. Third, the validation results in the Screening cohort (for outcomes with ≥ 20 cases) should be presented an Extended data table. Instead of a nominal p-value threshold you should apply a significance threshold either at an FDR rate or a Bonferroni threshold (0.05/31) to declare significance in validation. Fourth, there are 26 ICD-10 code

listings listed in Extended Data Table 3, rather than 20. Were some of these grouped together (if so, that should be made clear), or did some not reach 20 cases? Perhaps the number of cases should be listed in the Table?

We agree with the reviewer that we do not have a complete record of medical history for the individuals in our internal Biobank cohorts, since our outcomes are based on information available in the EHR system. While we are interested in studying age of onset for different diseases, the available information is the date of first diagnosis in our healthcare network. Other recent work has studied genetic associations with age of disease onset in biobank-scale cohorts, such as UK Biobank, with similar limitations (Staley 2017, Dey 2022), but we agree that there are significant challenges in accounting for the various forms of ascertainment in EHR and biobank studies, which merit comparing different approaches.

To account for left-truncation in our data, we have revised the survival models used in this section to handle delayed entry by explicitly using time from birth to the start of the observation period (defined as the first available diagnosis in EHR system) and the time from observation period to event or censoring. In addition, if subjects had a recorded relevant medical history prior to the start of the observation period or “instant events” where a diagnosis of interest occurred within 1 year of first healthcare encounter, we encoded the true event date as having occurred at an unknown timepoint in the interval between birth date and the start of the observation period.

When comparing the Cox models with delayed entry to our earlier approach (removing subjects with prior history, but no delayed entry), we obtained similar results, with typically more conservative estimates for the delayed entry models (R^2 of $-\log_{10} p$ - values 0.87, slope=0.83, intercept=0.08, correlation p -value $<1e-4$).

We have added a limitation section describing our time-to-event analyses in cohorts with EHR-based outcomes, including delayed entry as well as the difference between age-of-onset and age-of-diagnosis. Pg. 11 Line 19-25.

3. In general, those conditions that tested the largest sample size yielded the most significant results (Extended Data Table 1), whereas some of the drug stimuli, TMAO, etc. did not. It may reflect the variability in power among the tested conditions given the wide range in sample size and this may be worth comment in terms of potential False Negatives here, and the design of future studies toward larger samples. I think you should make some comment on the limitations of your varied power across the experiments since you have nearly a 10-fold range in sample sizes with different components.

We agree that the sample size and statistical power are different among different

perturbations. We have added discussions regarding this limitation. “... we had varying sample sizes across different perturbational conditions, which could lead to reduced statistical power for conditions with fewer subjects, potentially resulting in false negative findings.” Pg. 11 Line 9-11.

4. One of the problems that arises in flow cytometry is the migration of analytic gates across time which can be due to a variety of factors including changes in antibody batches/clones/manufacture, fluctuation in instruments requiring calibration and voltage settings, changes in detectors, fluctuating or inconsistent pressure within the system, etc. You mention that you initially tested the reproducibility of the various assays and conditions, but you do not present any of this data. Since this assay approach is relatively new knowledge in large PBMC samples, it would be helpful if you presented results on the reproducibility and other QC metrics in a Supplement. Do you have any repeated blood donors over the course of the study that might address the stability of the analytic gates within samples? You present a merged view of the data in a heatmap in Extended Data Figure 1. This is somewhat informative. There is also a movement in Flow cytometry to present multiple representative Gating plots from independent samples (akin to the prior movement to show several Western Blot images or other gel images). It might be helpful to show several independent samples gating for the various tests and conditions in a Supplement.

This is an important point and was also noted by Reviewer 1. We agree that flow cytometry data can exhibit batch effect affected by both reagents and calibration of the machine. However, one advantage of Sysmex is that it is a clinical hematology analyzer that has been engineered for high reproducibility at low cost. Importantly, we ran QC using a Sysmex QC kit every day at machine startup. We have now included these details in the Method section (Pg.12 Line 28). In addition, we have now included data showing replicates of blood measurements (under both baseline and one perturbed condition) for 4 randomly chosen exemplary subjects over time and from the same blood draw (new Extended data Figure 2A-B). The new Extended data Figure 2C shows several examples of calculated blood traits based on gating over time. This figure also confirms that our gating strategy and conditions remain stable and reproducible over time.

5. I am curious about several loci in Figure 2 that seem to show “pleiotropic” patterns across stimuli but are not commented on by the authors. These often seem to show associations with Baseline, Water, KCl, LPS, and/or other conditions. Why do you not focus at all on these – are they suspected to be generalized cellular QTLs that are not very condition- specific, or could they be loci that are sensitive and important for responses to many stimuli? What does a Water QTL represent – some sort of osmotic or pH change sensitive response?

Thank you for highlighting these features. We agree that many of the loci we identified are associated with multiple different blood traits. This may be a consequence of several factors. Firstly, to reduce false discovery, we only reported

clumps with at least two associated blood traits, which enriches for those loci that are “pleiotropic”.

Secondly, as you have pointed out, many of our blood traits are correlated with each other. One reason that may explain the shared genetic associations observed under Water, KCL, and LPS conditions is that they were all performed after prolonged incubation (over 14h). This extended incubation time itself can be considered a unique perturbation, which elicited blood cell responses that are common among these conditions, thus exhibiting shared genetic associations.

We have now added an estimation of independent traits based on principal component analyses, which revealed that we have at least 350 independent traits in our current dataset, though we measured thousands of primary blood traits. We have also added more discussion addressing this point (Extended Data Fig 5, Pg. 3, Pg. 11).

MINOR COMMENTS

1. The authors apply a standard/older GWAS threshold of $5E-08$ despite testing many traits. They also utilize an older imputation panel (1000G) than current panels such as HRC and TOPMed that provide better coverage and imputation. Personally, this is fine with me as a screening approach for further downstream uses (as in this case) [I think it would have been inadequate had the study just stopped there]. Other reviewers and readers may feel differently as the GWAS field has generally moved toward more conservative thresholds as # of variants tested and # of traits per study have both increased. Again, from my perspective this is not vital to address, but I think I would be remiss not to raise it as a potential concern.

Thank you for pointing this out. Due to our IRB protocol restrictions, we could not upload genetic data onto external services for imputation using the TOPMed panel. We instead used a local imputation server with the 1000G panel, which was publicly available.

We have now added a limitation section in our discussion regarding false positives. “... we utilized a conventional threshold of $5E-8$ for genetic association without adjusting for the number of tested phenotypes. While we tested thousands of phenotypes, many of those are correlated, and we estimated that approximately 350 traits were independent. However, this estimation may not be accurate, since it was based on a subset of subjects with the highest number of common perturbational conditions measured. To reduce the false discovery rate, we reported significant associations only when at least two independent traits were linked to the clumped region. Nevertheless, due to our relatively small sample size and approach, some false positive associations may still be present in the results.” Pg. 11 Line 1-8.

Importantly, the approaches we outline can be further scaled (much larger small molecule libraries) radically increasing the number of novel independent traits which can be accessed. In addition, we deliberately focused on human cellular phenotypes that can be readily exploited in direct experimentation and also directly translated to disease models. These features allow the use of iterative empiric biological validation, in addition to validation through genetic studies in independent cohorts.

2. Given recent publication on use of race and ancestry in genetics (e.g., NAS and TOPMed publication), please be more specific about how ancestry was defined in your sample.

In the perturbation screening cohort, we used self-reported race in the EHR record as well as self-reported values provided by participants during study entry. In the MGB Biobank cohort, plinkQC was used to determine ancestry based on a joint PCA projection with 1000G subjects. Samples with a distance greater than 3x radius of 1000G EUR reference samples were excluded from the PGS analyses. In the UKBB cohort the white ethnic background cohort based on the self-reported UKBB field f21000 was used for PGS analyses. We have now specified these details in our Method section. Pg. 13.

3. Is 39C a typo, or perturbations done above physiological temp for a reason? Please explain.

This is not a typo. We intentionally performed perturbational conditions at 39 °C, which was originally designed to simulate fever. However, we also observed that blood cells, especially neutrophils, survive longer under 39 °C compared to 37 °C, which extended the timeframe for us to perform the measurements.

4. Was it a single Sysmex instrument or multiple ones? If multiple, did you analyze possible instrument variability?

All measurements for the GWAS were done on a single Sysmex XN1000 instrument. Technical replication using a single machine for different blood draws is now assessed and shown in Extended Data Figure 2. We did not explore instrument variation across devices as there is extensive documentation of the low variation with this instrument as deployed in clinical laboratories worldwide.

5. Extended Table 3 – K85.0 code is listed twice, and unclear if it really differed from K85

We apologize for this mistake. We have now revised and harmonized the disease definitions for outcomes between MGB and UKBB. We have updated the Extended

Table 3 (which is the new Extended Table 4) to include ICD10 code prefix and UKBB trait IDs.

6. What measure of LD >0.50 for clumping (R^2 , D , D')?

We apologize for the omission of the unit for LD. We have extended the description of the clumping step to include the full plink command: “we clumped all significant variants using plink with LD $r^2 >0.50$, physical distance $<250\text{kb}$ between clumped variants, and at least two independent hits from different traits for each clumped region. We used the variant with the smallest association p-value across all measured traits for a given region as the lead variant. The following command was used for clumping and gene range annotations: `plink --clump-range glist-hg19 --clump-p1 0.00000005 --clump-p2 0.00000005 --clump-r2 0.50 --clump-kb 250 --clump-replicate --clump {trait_files}`.” Pg. 17, Line 20-26.

7. Please define what padj means at first use (i.e., FDR threshold or whatever it is if it varies across analyses)

Thank you for pointing out this omission. We have added the following description of the multiple-testing adjustment to the sections on associations with clinical outcomes:

“p-values were adjusted for false discovery rate (FDR) to account for multiple testing across 327 perturbation-blood responses and 50 clinical outcomes, including 20 lab values and 30 diagnoses.” Pg.4, Line 40-42.

“Points indicate significant associations after multiple testing correction using FDR across all tested diseases, lab values and blood traits (50 clinical outcomes and 327 blood readouts) with adjusted p-value thresholds: ... ” Figure 3 Legend, Pg. 23, Line 11-13.

“Points indicate significant associations after multiple testing correction across all tested diseases and blood traits (30 clinical outcomes and 327 blood readouts) with adjusted p-value thresholds: ... ” Figure 7 Legend, Pg. 29, Line 12-14.

8. In Extended Data Fig. 7 for RBC trait, no data was plotted in my version for Age vs. RBC1 Med SFL. I’m not sure if this is an author omission or a problem with the journal processing or PDF download on my end.

We apologize that there was a problem with the plot while converting. We have now fixed this figure.

9. In the Introduction you “utilized live circulating” cells. This is not exactly accurate. You

removed the cells from circulation before assaying as is typically the case for most such assays (unless in vivo imaging is used, or a proxy circulatory system used). So I suggest modifying the phrase slightly here.

We have corrected this statement to “live human blood cells”.

10. In the Introductory description, which is overall very well written, in circulation you fail to mention cell-free DNA, microparticles and extra-cellular vesicles which are other mechanisms for biological readouts or cell-cell communication.

Thank you for the suggestion. We have added those factors in the sentence. Pg. 2, Line 13-14.

11. In addition to the HFGP project, there is a corollary study type which is probing platelet functional responses to agonist stimuli (the largest/most recently WGS in $n \sim 3,800$ Keramati et al. Nat Commun PMID 34131117 and Rodriguez et al. AJHG PMID 32649856 which found a large QTL associated with venous and arterial thrombosis).

We have now cited these two studies in the introduction. Pg. 2, Line 26-27.

12. Ref. 19 has been on BioRxiv server for 3+ years. I'm not sure if there is an age where a preprint becomes less appropriate to cite but this seems old to me. Perhaps this is a journal question, or perhaps the article will be published and citation changed by the time this article might go to press.

We kept this BioRxiv citation due to its relevance to our work and as it is still not published as a peer-review article at the time of our revision.

Reviewer #3:

Remarks to the Author:

The authors utilize an innovative GWAS-based framework to study various endophenotypes related to a broad range of chemical and pharmacologic perturbations of blood cells measured using a routine hematology analyzer. The combinations of cell types, readouts, fluorophore, and ~ 40 chemical stimuli generated several thousand parameters per subject in up to 3300 subjects. Using this approach, 100 loci were associated with at least two blood cell-perturbation readouts, including 23 perturbation loci not previously reported. The authors then performed focused investigation and further characterization of a particular subpopulation (NE2) of apoptotic neutrophils that appear following inflammatory stimuli whose presence is inversely related to extent of neutrophil activation and pro-inflammatory response and also inversely

associated with risk of clinical cardiometabolic phenotypes in the same MGB data set. These included experiments on neutrophil metabolism and zebrafish related to candidate genes HK1 and PFKP. The authors also generated polygenic scores for the evoked blood cell endophenotypes and assessed their relationship to clinical outcomes in MGB.

We thank the reviewer for their valuable feedback. We have addressed each comment in detail below and incorporated the suggested improvements to our manuscript.

Comments:

1. The subjects were recruited from phlebotomy clinics of a tertiary hospital system, and therefore may be enriched for specific conditions or diagnoses, rather than being representative of the general population. Therefore, it would be useful to describe or summarize the clinical, demographic, and lifestyle characteristics of the study participants. Were individuals excluded for conditions known to strongly influence blood cell parameters (e.g., primary hematologic or bone marrow disorders, cancer, myelosuppressive chemotherapy) or acute inflammatory conditions? This would be important to assess how generalizable the findings might be to the general population.

We agree with the reviewer that our study cohort consists of individuals with a higher prevalence of medical conditions compared to the general population. We did not exclude individuals with specified diseases, because we did not want to arbitrarily bias our sampling. However, we did exclude a small number of subjects with extremely abnormal baseline blood count.

To address the potential bias in our study cohort and validate our findings in a more general population, we have now calculated PGS using a meta-analysis of both the MGB cohort and a more general UK Biobank cohort (new Figure 7). Our results showed that the blood endophenotype-based PGSs exhibited significant stratification of numerous key diseases, consistently evident in both our MGB cohort and the UK Biobank cohort (new Figure 7A and B). This concordance between the two cohorts suggests that our findings can be applied to a broader population.

However, we also noted that there were disease associations that were only present in the MGB cohort or the UK Biobank cohort (Extended Data Figs 10 and 11), possibly arising from differences between these two datasets. The MGB cohort benefits from more precise disease definitions and fine-grained diagnostics/clinical measurements,

while in the UK Biobank, many of the disease traits were self-reported. Additionally, the MGB dataset consists of many diseases that are not defined in the UK Biobank. Conversely, the significantly larger sample size of the UK Biobank allowed for the

identification of many significant disease associations that were not observed in the MGB cohort (Extended Data Figs. 10 and 11). We included this discussion on Pg. 9, Line 4-11.

2. Fig 1D – what is the X-axis?

We have updated Fig. 1D and its corresponding Fig. 3C to show the independent component analysis of blood trait associations with blood traits and updated the results. “The ICA analysis effectively grouped clinical endpoints and lab values into meaningful clusters, revealing, for instance, a cluster encompassing obesity, T2D, and glucose measurements, as well as another cluster comprising asthma, chronic obstructive pulmonary disease, and venous thrombosis (Fig. 3C). Additionally, we plotted the representation of seven example blood traits into the same IC space (Fig. 3C, arrows), demonstrating how each blood trait carries unique information related to various clinical phenotypes.” Pg. 5 Line 9-14.

3. It is stated that non-European individuals were excluded from GWAS. What is the race/ethnicity breakdown of the participants? How many non-European individuals were excluded? Is the final # included in the GWAS = 2685? Given that there are a known differences in the genetic architecture of blood cell counts and inflammatory responses between different race/ethnicities, how generalizable are the findings expected to be to non- Europeans?

The race/ethnicity breakdown of all participants we profiled is shown in Extended Table 3. Notably, we used the evoked blood cell traits from all ancestries for the clinical/phenotypical associations. For the GWAS, we first focused on subjects of European ancestry (we have now specified the numbers of European ancestry individuals passing QC for each condition in the updated Extended Data Table 1), as our number of individuals of other ancestry groups was not sufficient for robust GWAS discovery (the total number of European ancestry subjects passing QC is 1940). However, we have now performed GWAS validation on other ancestries for selected perturbational conditions and included a new supplementary figure illustrating the beta coefficient and p-values for multi-ancestry associations of several lead SNPs (Extended Data Figure 13). We discussed this result and limitation in the text “...Furthermore, while our phenotypic associations are based on all ancestry groups, the genetic association findings are based on individuals of European ancestry due to the limited representation of subjects from other ancestries in our screening cohort. We performed GWAS analyses for a subset of blood cellular traits across multiple ancestry groups, which revealed consistent trends in effect directions, albeit with notable disparities among various ancestral groups for several lead SNPs (Extended Data Figure 13). Future investigations encompassing a larger cohort of individuals with non-European backgrounds are needed to unravel the trans-ancestry genetic basis governing evoked blood cellular

responses.” Pg.11 Line 12-19.

We also intend to undertake similar studies in non-Europeans in the future, which has been made feasible by the widespread availability of the Sysmex technology and the low cost of our perturbational assays.

4. The conventional genome-wide significance threshold of 5×10^{-8} seems liberal, given that thousands of different phenotypes (albeit some of them correlated) were tested. The effective number of uncorrelated traits could be estimated and used to correct for multiple testing.

We have now added a limitation section in our discussion regarding this point and added an estimation for the number of uncorrelated traits. “... we utilized a conventional threshold of $5E-8$ for genetic association without adjusting for the number of tested phenotypes. While we tested thousands of phenotypes, many of those are correlated, and we estimated that approximately 350 traits were independent. However, this estimation may not be accurate, since it was based on a subset of subjects with the highest number of common perturbational conditions measured. To reduce the false discovery rate, we reported significant associations only when at least two independent traits were linked to the clumped region. Nevertheless, due to our relatively small sample size and approach, some false positive associations may still be present in the results.” Pg. 11 Line 1-8. We plan to continue validating additional associations experimentally given the specificity of the cellular phenotypes which we have uncovered.

5. I am quite confused about the description of the Results related to Extended Data Fig 5. The paragraph discusses an increase in NE2/NE4 ratio with Pam3CSK4 treatment and the relationship of higher NE2/NE4 to neutrophil proinflammatory responses. Which appears to be the exact opposite of what the data in the Figure shows.

Thank you for pointing out this error in our description of these results. We have now corrected this (Pg. 6, Line 36-39). Pam3CSK4 decreased NE2/NE4 ratio and increased neutrophil pro-inflammatory responses.

6. The authors state “The lead SNPs we identified cause upregulation in HK1 and PFKP expression and decreased NE2/NE4 ratio, suggesting reduced neutrophil apoptosis.” What are the data that suggest that these SNPs cause upregulation of HK1 and PFKP?

We apologize for not including the references for this statement. We stated this based on the eQTL of the lead SNPs for these two genes. We have now included the references for the eQTL data. “The lead SNPs we identified for HK1 and PFKP were

previously found to cause upregulation in these two gene expressions (rs6480404 eQTL for HK1 in neutrophil (Chen et al., Cell, 2016): $\beta=0.178$, $p=4e-16$; rs34538474 eQTL for PFKP in blood: $\beta=0.457$, $p=3.3e-310$) (Vösa & Claringbould et. al., Nature Genetics, 2021).” Pg. 7 Line 5-7.

7. The authors report that polygenic scores for evoked blood cell readouts predict occurrence of various clinical outcomes within the same MGB biobank data set. Since only genotype data are required to assess the association of PGS with clinical outcomes, it would be helpful to see some of these PGS – clinical trait associations replicated in independent data sets with genome-wide genotyping available.

We agree that this is a useful approach to validation, although there are disparities between trait definitions EHR and population cohorts. We have now included the UK Biobank dataset for the PGS and clinical trait analysis. We performed a meta-analysis of MGB and UK Biobank cohorts (New Fig.7), as well as calculated PGS using each of these cohorts (New Extended Data Fig. 10 and 11). The meta-analysis showed that the PGS successfully stratified the onset of many diseases in both cohorts. “...for example, obesity (ret LPS 18h RBC2 Med SSC, $p_{adj}=3.74E-06$, MGB cases=9499, UKBB cases= 41893), Type 2 diabetes (ret KCI 17h RET1 %, $p_{adj}=1.5E-4$, MGB cases=6226, UKBB cases=34941), chronic kidney disease (wnr Water 15h WBC2 Med FSC, $p_{adj}=1.7E-05$, MGB cases=5627, UKBB cases=23771), and heart failure (ret KCI 17h RBC1 SD FSC, $p_{adj}=6.4E-3$, MGB cases=4421, UKBB cases=15811). In addition, we also observed strong associations with immune system conditions such as T1D (pltf LPS 18h PLT F Med SFL $p_{adj}=8.6E-05$, MGB cases=530 , UKBB cases=4207), asthma (wnr LPS 18h WBC Med SSD, $p_{adj}=8.7E-05$, MGB cases=6176, UKBB cases=62009) and systemic lupus erythematosus (wdf Alhydrogel 21h NE2-NE4 ratio, $p_{adj}=4.5E-03$, MGB cases=532, UKBB cases=804).” The detailed results are explained on Pg. 8-9.

Decision Letter, first revision:

21st September 2023

Dear Dr. Homilius,

Your revised manuscript "Perturbational phenotyping of human blood cells reveals genetically determined latent traits associated with subsets of common diseases" (NG-A62115R) has been seen by the original referees. As you will see from their comments below, the referees are satisfied with the revision and have no remaining requests, and therefore we will be happy in principle to publish your study in Nature Genetics as an Article pending final revisions to comply with our editorial and formatting guidelines.

We are now performing detailed checks on your paper, and we will send you a checklist detailing our editorial and formatting requirements soon. Please do not upload the final materials or make any revisions until you receive this additional information from us.

Thank you again for your interest in Nature Genetics. Please do not hesitate to contact me if you have any questions.

Sincerely,
Kyle

Kyle Vogan, PhD
Senior Editor
Nature Genetics
<https://orcid.org/0000-0001-9565-9665>

Reviewer #1 (Remarks to the Author):

I don't have additional comments.

Reviewer #2 (Remarks to the Author):

In my opinion the authors have adequately addressed all the prior peer review comments. On the whole the work is an excellent demonstration of large-scale perturbation QTLs strengthened by additional functional follow-up and clinical associations.

Reviewer #3 (Remarks to the Author):

The authors have satisfactorily addressed my concerns.

Final Decision Letter:

27th October 2023

Dear Dr. Homilius,

I am delighted to say that your manuscript "Perturbational phenotyping of human blood cells reveals genetically determined latent traits associated with subsets of common diseases" has been accepted for publication in an upcoming issue of Nature Genetics.

Your paper will be published online after we receive your corrections and will appear in print in the next available issue. You can find out your date of online publication by contacting the Nature Press Office (press@nature.com) after sending your e-proof corrections. Now is the time to inform your Public Relations or Press Office about your paper, as they might be interested in promoting its publication. This will allow them time to prepare an accurate and satisfactory press release. Include your manuscript tracking number (NG-A62115R1) and the name of the journal, which they will need when they contact our Press Office.

Before your paper is published online, we will be distributing a press release to news organizations worldwide, which may very well include details of your work. We are happy for your institution or funding agency to prepare its own press release, but it must mention the embargo date and Nature Genetics. Our Press Office may contact you closer to the time of publication, but if you or your Press Office have any enquiries in the meantime, please contact press@nature.com.

Please note that Nature Genetics is a Transformative Journal (TJ). Authors may publish their research with us through the traditional subscription access route or make their paper immediately open access through payment of an article-processing charge (APC). Authors will not be required to make a final decision about access to their article until it has been accepted. [Find out more about Transformative Journals](https://www.springernature.com/gp/open-research/transformative-journals)

Authors may need to take specific actions to achieve [compliance](https://www.springernature.com/gp/open-research/funding/policy-compliance-faqs) with funder and institutional open access mandates. If your research is supported by a funder that requires immediate open access (e.g. according to [Plan S principles](https://www.springernature.com/gp/open-research/plan-s-compliance)), then you should select the gold OA route, and we will direct you to the compliant route where possible. For authors selecting the subscription publication route, the journal's standard licensing terms will need to be accepted, including [self-archiving-and-license-to-publish](https://www.nature.com/nature-portfolio/editorial-policies/self-archiving-and-license-to-publish). Those licensing terms will supersede any other terms that the author or any third party may assert apply to any version of the manuscript.

If you have not already done so, we invite you to upload the step-by-step protocols used in this manuscript to the Protocols Exchange, part of our on-line web resource, natureprotocols.com. If you complete the upload by the time you receive your manuscript proofs, we can insert links in your article that lead directly to the protocol details. Your protocol will be made freely available upon publication of your paper. By participating in natureprotocols.com, you are enabling researchers to more readily reproduce or adapt the methodology you use. [Natureprotocols.com](http://natureprotocols.com) is fully searchable, providing your protocols and paper with increased utility and visibility. Please submit your protocol to <https://protocolexchange.researchsquare.com/>. After entering your [nature.com](http://www.nature.com) username and password you will need to enter your manuscript number (NG-A62115R1). Further information can be found at <https://www.nature.com/nature-portfolio/editorial-policies/reporting-standards#protocols>

Sincerely,
Kyle

Kyle Vogan, PhD
Senior Editor
Nature Genetics
<https://orcid.org/0000-0001-9565-9665>